EMBO
Molecular Medicine

# TDP-43 pathology triggers neuroinflammation and cognitive impairment by inducing microglial necroptosis

Shenrui Guo[1,4], Hongfu Jin[1,4], Hui Sun[1], Shuo Huang[1], Yuanyuan Chen[2], Yuge Chang[1], Yu Zhang[3], Lin Ding[1], Suyun Chen[1], Chenglai Fu [2✉], Yafu Yin [1✉] & Weiwei Cheng [1✉]

## Abstract

**Pathological TAR DNA-binding protein-43 (TDP-43) is a defining feature of several neurodegenerative diseases, including amyotrophic lateral sclerosis (ALS), frontotemporal dementia (FTD) and Alzheimer's disease (AD). However, the mechanism by which TDP-43 pathology disrupts microglial function and drives neuroinflammation remains unclear. In this study, we demonstrated that cytoplasmically mis-localized TDP-43 exacerbated neuroinflammation, induced cell death, and impaired phagocytic function in microglial cells, primarily through receptor interacting serine/threonine kinase 3 (RIPK3)-dependent necroptosis. Pharmacological inhibition of RIPK3 with GSK872 markedly attenuated these pathological effects in vitro. These findings were further corroborated in a murine model with cytoplasmic TDP-43 mis-localization, where GSK872 treatment remarkably alleviated neuroinflammation and restored cognitive deficits. Mechanistically, our findings indicate that the nuclear depletion of TDP-43, resulted from its cytoplasmic mis-localization, impairs its ability to transcriptionally repress the *Ripk3* gene, subsequently leading to RIPK3 upregulation and activation of RIPK3-dependent necroptosis. Collectively, our findings establish RIPK3-dependent necroptosis as a critical driver of TDP-43 pathology-mediated neuroinflammation and identified necroptosis as a promising therapeutic target in TDP-43-associated neurodegenerative disorders.**

**Keywords** Microglia; Necroptosis; Neuroinflammation; RIPK3; TDP-43
**Subject Categories** Autophagy & Cell Death; Neuroscience

## Introduction

Pathological TAR DNA-binding protein-43 (TDP-43) is a hallmark of neurodegenerative diseases, which exhibits characteristic hyperphosphorylation, ubiquitination, and truncation, resulting in cytoplasmic aggregation accompanied by nuclear depletion (Amador-Ortiz et al, 2007; Geser et al, 2008; Nakashima-Yasuda et al, 2007; Neumann et al, 2006). Although extensive research has been conducted on neuronal TDP-43 pathology, investigations into glial TDP-43 pathology have been relatively limited, primarily concentrating on the mechanisms by which TDP-43 pathology induces neuronal injury (Paolicelli et al, 2017; Zhang et al, 2008). The role of TDP-43 in microglia has not been thoroughly examined. Importantly, microglia-mediated neuroinflammation is a significant contributor to neurodegeneration, yet how TDP-43 pathology dysregulates microglia function is unknown.

Microglia, the resident immune cells of the central nervous system (CNS), exert pivotal roles in key neuroprotective processes including immune surveillance, homeostasis maintenance, injury repair, and phagocytosis (Gao et al, 2023). Among these functions, phagocytosis is particularly critical, as it facilitates the clearance of cellular debris and pathological protein aggregates, while also contributing to the regulation of synaptic plasticity (Paolicelli et al, 2011). Research demonstrates that microglia engulf degenerating motor neurons and intraneuronal TDP-43 aggregates, thereby mediating a neuroprotective effect (Svahn et al, 2018). Conversely, impaired phagocytic function can lead to the accumulation of neurotoxic aggregates and neuronal loss, a process that exacerbates the progression of neurodegenerative diseases (Hassan et al, 2025). Furthermore, the specific interaction between TDP-43 and the microglial receptor TREM2 enhances phagocytic clearance of TDP-43 aggregates, highlighting a crucial molecular pathway in microglial response to TDP-43 pathology (Xie et al, 2022).

However, under chronic pathological conditions such as those in TDP-43 proteinopathies, persistent stimulation can trigger a shift in microglial phenotype toward neurotoxic activation. Indeed, widespread microglial activation has been well documented in patients with TDP-43 proteinopathies such as amyotrophic lateral sclerosis (ALS) and frontotemporal dementia (FTD) (Cagnin et al, 2004; Togawa et al, 2024; Turner et al, 2004; Zürcher et al, 2015). An in vitro study has demonstrated that extracellular forms of TDP-43—encompassing wild-type, mutant, and truncated fragments—act as a potent microglia activator, initiating a proinflammatory cascade characterized by

---

[1]Department of Nuclear Medicine, Xinhua Hospital Affiliated to Shanghai Jiao Tong University School of Medicine, 200092 Shanghai, China. [2]Institute for Developmental and Regenerative Cardiovascular Medicine, Xinhua Hospital Affiliated to Shanghai Jiao Tong University School of Medicine, 200092 Shanghai, China. [3]Department of Neurology, Xinhua Hospital Affiliated to Shanghai Jiao Tong University School of Medicine, 200092 Shanghai, China. [4]These authors contributed equally: Shenrui Guo, Hongfu Jin.
✉E-mail: fuchenglai@xinhuamed.com.cn; yinyafu@shsmu.edu.cn; chengweiwei8416@xinhuamed.com.cn

cytokine/chemokine release that is directly toxic to motor neurons (Zhao et al, 2015). Additionally, the internalization of TDP-43 aggregates by microglia can lead to aberrant mobilization of endogenous microglial TDP-43, which may propagate neuroinflammation (Leal-Lasarte et al, 2017). Critically, impaired phagocytosis was observed in monocyte-derived microglia-like cells from ALS patients, which themselves contain both non-phosphorylated and phosphorylated-TDP-43-positive inclusions (Quek et al, 2022). Paradoxically, however, TDP-43 depletion in microglia has also been associated with a hyperphagocytic state that promotes amyloid clearance but simultaneously induces significant synapse loss (Paolicelli et al, 2017), and contributes to neuronal death (Xia et al, 2015). Collectively, this phagocytic dysfunction within diseased microglia likely represents a key mechanism contributing to the accumulation of toxic aggregates and disease progression.

Our study established that mis-localized TDP-43 exacerbated microglia-mediated neuroinflammation primarily through the induction of necroptosis. Necroptosis, a pro-inflammatory form of programmed cell death implicated in neuroinflammation (Yuan et al, 2019), has been observed in patients with multiple neurodegenerative diseases including Alzheimer's disease (AD), multiple sclerosis (MS), and Parkinson disease (PD) (Caccamo et al, 2017; Ofengeim et al, 2015; Oñate et al, 2020). In this study, we demonstrated that mis-localized TDP-43 loses its nuclear function in transcriptionally inhibiting receptor interacting serine/threonine kinase 3 (Ripk3) gene, thereby leading to RIPK3 upregulation and triggering RIPK3-dependent necroptosis in microglia. These findings identified RIPK3 as a promising therapeutic target for alleviating neuroinflammation, microglial cell death, and disease progression in neurodegenerative disease.

# Results

## TDP-43 NLSmut induced neuroinflammation, cell death and phagocytosis inhibition in BV2 microglial cells

To investigate the pathological effects of TDP-43 mis-localization in microglia, we established BV2 microglial cells stably expressing either GFP (BV2-GFP) or nuclear localization sequence-mutated TDP-43 (BV2-NLSmut) via lentiviral infection. Consistent with previous study (Winton et al, 2008), overexpression of TDP-43 NLSmut in BV2 microglial cells resulted in cytoplasmic accumulation of TDP-43 and decreased nuclear levels of endogenous TDP-43 protein (Figs. 1A,B and EV1A). Compared to BV2-GFP control cells, significant upregulation of phosphorylated TDP-43 and truncated TDP-43 C-terminal fragments (CTFs) were observed, as demonstrated by western blot bands at 37 kDa and 25 kDa, while full-length TDP-43 levels remained unchanged (Fig. 1C).

This pathological TDP-43 profile correlated with microglial activation, as evidenced by elevated Iba1 protein expression in BV2-NLSmut cells (Fig. 1D). Subsequent ELISA quantification demonstrated a significant increase in the secretion of pro-inflammatory cytokine TNF-α, IL-1β and IL-6 in BV2-NLSmut cells (Fig. 1E), while qPCR analysis confirmed corresponding upregulation of Tnf, Il1b, Il6 mRNA levels in BV2-NLSmut cells (Fig. EV1B). Additionally, the accumulation of intracellular TNF-α protein was validated by western blot analysis (Fig. EV1C). These results collectively suggest that TDP-43 mis-localization triggered a robust neuroinflammatory response in BV2 microglial cells. Furthermore, CCK-8 assay showed a significant reduction in cell viability in BV2-NLSmut cells compared to BV2-GFP control cells (Fig. 1F). Phagocytosis assay with latex beads revealed significantly less beads engulfed by BV2-NLSmut cells in comparison to BV2-GFP control cells, indicating a diminished phagocytic function in BV2-NLSmut cells (Fig. 1G).

## TDP-43 NLSmut activated necroptosis in BV2 microglial cells

To investigate the molecular mechanisms underlying TDP-43 mis-localization induced neuroinflammation in BV2 microglial cells, we conducted transcriptomic profiling of BV2-NLSmut cells. RNA-seq analysis identified 1432 differentially expressed genes (DEGs), comprising 909 upregulated and 523 downregulated genes compared to BV2-GFP controls, by using cutoff criteria of adjusted $p$ value < 0.05 and fold change>2 ($|Log2FC| > 1$) (Fig. 2A). The top 50 most significantly upregulated and downregulated DEGs in BV2-NLSmut cells compared to BV2-GFP control cells, ranked by fold change, were presented in Datasets EV1 and 2. qPCR validation of selected DEGs confirmed the reliability of transcriptomic data (Appendix Fig. S1). Gene Ontology (GO) enrichment analysis revealed significant enrichment of inflammatory response pathway (Fig. 2B).

Notably, RNA-seq data revealed a significant upregulation of RNA level of Ripk3, a crucial regulator of necroptosis, in BV2-NLSmut cells compared to BV2-GFP control cells. This observation was further validated by qPCR, showing more than 40-fold robust increase in Ripk3 mRNA levels (Fig. 2C). Western blot analysis further demonstrated a substantial elevation in both RIPK3 protein expression and its phosphorylation, along with increased phosphorylation of its downstream effector, mixed lineage kinase domain-like protein (MLKL) (Fig. 2D). Immunofluorescence also shown elevated RIPK3 protein expression in BV2-NLSmut cells compared to BV2-GFP control cells (Fig. 2E). MLKL protein is the pivotal component in the necroptosis pathway; upon activation via phosphorylation by RIPK3, MLKL undergoes a conformational change and translocates to the plasma membrane, facilitating membrane permeabilization and ultimately leading to cell death (Chen et al, 2014). The increase of cell membrane permeability results in the release of intracellular lactate dehydrogenase (LDH) into the extracellular space, which can serve as an indicator of cell injury or cytotoxicity. Therefore, we performed LDH release assays and observed a notable rise in extracellular LDH activity, suggesting plasma membrane damage in BV2-NLSmut cells (Fig. 2F). Furthermore, immunofluorescence co-staining with the cellular membrane marker DiD confirmed the membrane translocation of phosphorylated MLKL (Fig. 2G). All these findings establish necroptosis pathway activation as a critical mechanism in TDP-43 mis-localized BV2 microglial cells.

## Necroptosis activation was essential for TDP-43 NLSmut induced neuroinflammation, cell death and phagocytosis inhibition

Next, we pharmacologically inhibited necroptosis activation with GSK872, a specific inhibitor of RIPK3 which inhibits its kinase activity (Mandal et al, 2014), to ascertain whether RIPK3

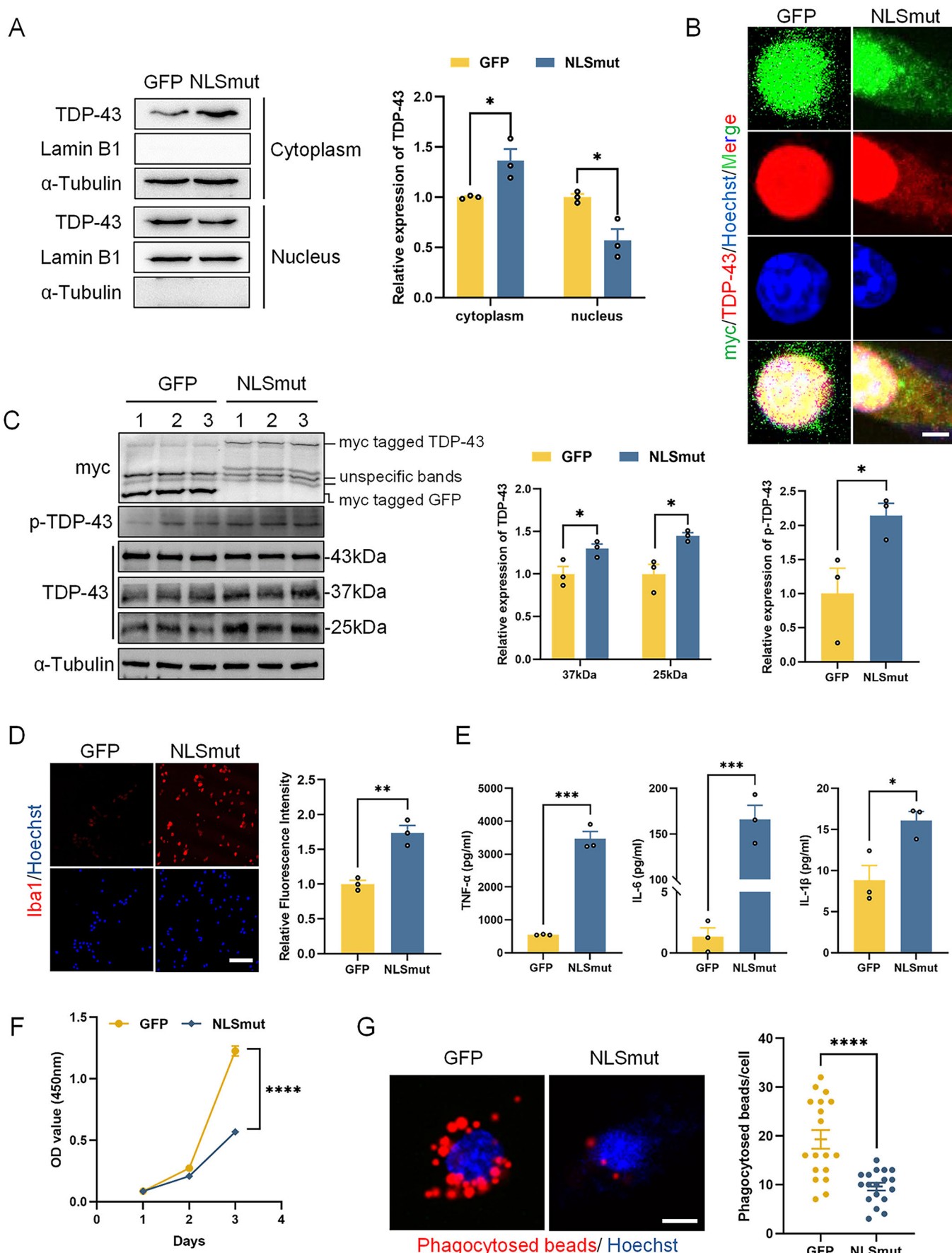

**Figure 1.  TDP-43 NLSmut induced neuroinflammation, cell death and phagocytosis inhibition in BV2 microglial cells.**

(A) Immunoblotting using antibody recognizing TDP-43, Lamin B1 and α-Tubulin in either cytoplasmic or nucleus fraction from BV2-GFP and BV2-NLSmut cells, along with the corresponding quantitative analysis. α-Tubulin was used as the cytoplasm loading control, Lamin B1 was used as the nucleus loading control. Data are presented as mean ± SEM from three biological replicates and analyzed by unpaired two-tailed Student's t test. For cytoplasm, p = 0.0345, for nucleus, p = 0.0213. (B) Immunofluorescence of myc (green) and TDP-43 (red) in BV2-GFP and BV2-NLSmut cells. Hoechst (blue) for nuclei. Scale bar, 5 μm. (C) Immunoblotting using antibody recognizing myc tag, phospho-TDP-43 (p-TDP-43) at the Ser409/410 site (p = 0.0499), TDP-43 (for 37 kDa CTF, p = 0.0469; for 25 kDa CTF, p = 0.0189), and α-Tubulin in BV2-GFP and BV2-NLSmut cells, along with the corresponding quantitative analysis. Data are presented as mean ± SEM from 3 biological replicates and analyzed by unpaired two-tailed Student's t test. (D) Immunofluorescence and the corresponding quantification of Iba1 (red) in BV2-GFP and BV2-NLSmut cells. Hoechst (blue) for nuclei. Scale bar, 100 μm. Three fields of view were analyzed for fluorescent intensities. Data are mean ± SEM and analyzed by unpaired two-tailed Student's t test. p = 0.0033. (E) ELISA analysis of TNF-α (p = 0.0002), IL-6 (p = 0.0004) and IL-1β (p = 0.0272) secreted by BV2-GFP and BV2-NLSmut cells. Data are mean ± SEM from three biological replicates and analyzed by unpaired two-tailed Student's t test. (F) Cell viability of BV2-GFP and BV2-NLSmut cells measured by CCK-8 assay. p = 0.00000003. Data are mean ± SEM from six biological replicates and analyzed by unpaired two-tailed Student's t test. (G) Immunofluorescence and the corresponding quantification indicating phagocytosed beads (red) in BV2-GFP and BV2-NLSmut cells. Hoechst (blue) for nuclei. Scale bar, 5 μm. At least 15 cells of each group were analyzed. Data are mean ± SEM and analyzed by unpaired two-tailed Student's t test. p = 0.00005. *p < 0.05, **p < 0.01, ***p < 0.001, ****p < 0.0001. Source data are available online for this figure.

upregulation and the subsequent necroptosis activation are essential to mediate the pathological phenotypes of microglial BV2 cells including increased neuroinflammation, reduced cell viability and phagocytic function. GSK872 treatment markedly suppressed the phosphorylation of RIPK3 and MLKL in BV2-NLSmut cells (Fig. 3A), and it led to significantly decreased LDH release and reduced translocation of phosphorylated MLKL to the plasma membrane in these cells (Fig. 3B,C). Immunofluorescence experiments showed that GSK872 treatment significantly diminished Iba1 expression in BV2-NLSmut cells (Fig. 3D), suggesting an amelioration in neuroinflammation. Meanwhile, treatment with GSK872 resulted in a notable reduction in the secretion of pro-inflammatory cytokines TNF-α, IL-1β, IL-6 by BV2-NLSmut cells (Fig. 3E), a decrease of intracellular TNF-α expression (Fig. EV2B), and a significant downregulation of the mRNA expression levels of these cytokines (Fig. EV2A). A marked restoration of cellular viability and phagocytic ability by GSK872 was also observed in BV2-NLSmut cells (Fig. 3F,G).

In addition to the pharmacological approach, we genetically inhibited RIPK3 through shRNA mediated knockdown to investigate its essential function in BV2 cells with TDP-43 mis-localization. Western blot analysis and IF staining revealed that the knockdown of RIPK3 led to a substantial reduction in the levels of phosphorylated RIPK3 and phosphorylated MLKL, as well as the translocation of phosphorylated MLKL to the plasma membrane (Fig. 4A,B). Moreover, the cell viability and cell phagocytic capacity was significantly improved by RIPK3 knockdown in TDP-43 mis-localized BV2 microglial cells (Fig. 4C,D). These findings conclusively established that RIPK3 mediated necroptosis serves as the critical mechanism by which TDP-43 mis-localization leads to BV2 microglial dysfunction, integrating neuroinflammatory activation with reduced cellular viability and decreased phagocytosis function.

## Decreased nuclear TDP-43 expression enhanced *Ripk3* transcription and induced necroptosis

As previously noted, we observed a robust upregulation of *Ripk3* mRNA in BV2 microglial cells with mis-localized TDP-43 (Fig. 2C). To elucidate the molecular mechanisms responsible for the upregulation of *Ripk3*, we first aimed to determine whether the upregulation was due to enhanced transcription or reduced mRNA

degradation. Utilizing Actinomycin D to inhibit new transcripts production, we found that *Ripk3* mRNA decay kinetics were comparable between BV2-GFP and BV2-NLSmut cells (Fig. 5A), thereby excluding post-transcriptional regulation as a contributing factor for *Ripk3* mRNA upregulation. Importantly, the quantification of nascent transcripts revealed significantly elevated levels of *Ripk3* pre-mRNA in BV2-NLSmut cells (Appendix Fig. S2A), with cellular RNA fractionation confirming a preferential accumulation of un-spliced precursors within the nuclear compartment (Appendix Fig. S2B). These findings indicated that the elevated *Ripk3* mRNA levels are primarily due to enhanced transcriptional activity.

TDP-43 is a crucial regulator of transcription and has been identified as a transcription repressor in many contexts (Ayala et al, 2008; Lalmansingh et al, 2011; Schwenk et al, 2016). We therefore hypothesized that the nuclear depletion of TDP-43, resulting from its mis-localization to the cytoplasm, disrupts its inherent transcriptional repressive function at the *Ripk3* locus. The consequent loss of repression drives de novo transcription of *Ripk3*, leading to increased expression of total RIPK3 and phosphorylated RIPK3, thereby propagating necroptotic signaling. We further validated enhanced transcription of *Ripk3* and the induction of necroptosis in BV2 microglial cells with diminished nuclear TDP-43 expression through shRNA-mediated knockdown (Fig. 5B). Consistent with what we observed in BV2-NLSmut cells, cells with nuclear TDP-43 knockdown recapitulated the necroptotic activation—characterized by concomitant upregulation of RIPK3 expression and its phosphorylation (Fig. 5C,D), along with the phosphorylation of the downstream effector MLKL (Fig. 5D)—revealing a conserved mechanism of necroptosis induction associated with the depletion of nuclear TDP-43.

## TDP-43 repressed *Ripk3* transcription via RRM1 domain binding to the (GT)₁₇ sequence in *Ripk3* gene promoter

Given the established role of TDP-43 as a transcriptional repressor reliant on its RNA recognition motif 1 (RRM1) domain (Lalmansingh et al, 2011), we investigated its regulatory interaction with RNA polymerase II at the *Ripk3* locus. Chromatin immunoprecipitation (ChIP) assays revealed a decrease in TDP-43 binding at the promoter regions of *Ripk3*, specifically at sites of −1.0 kb, −0.2 kb and +0.1 kb (Fig. 5E,F). Conversely, the binding of Rpb1,

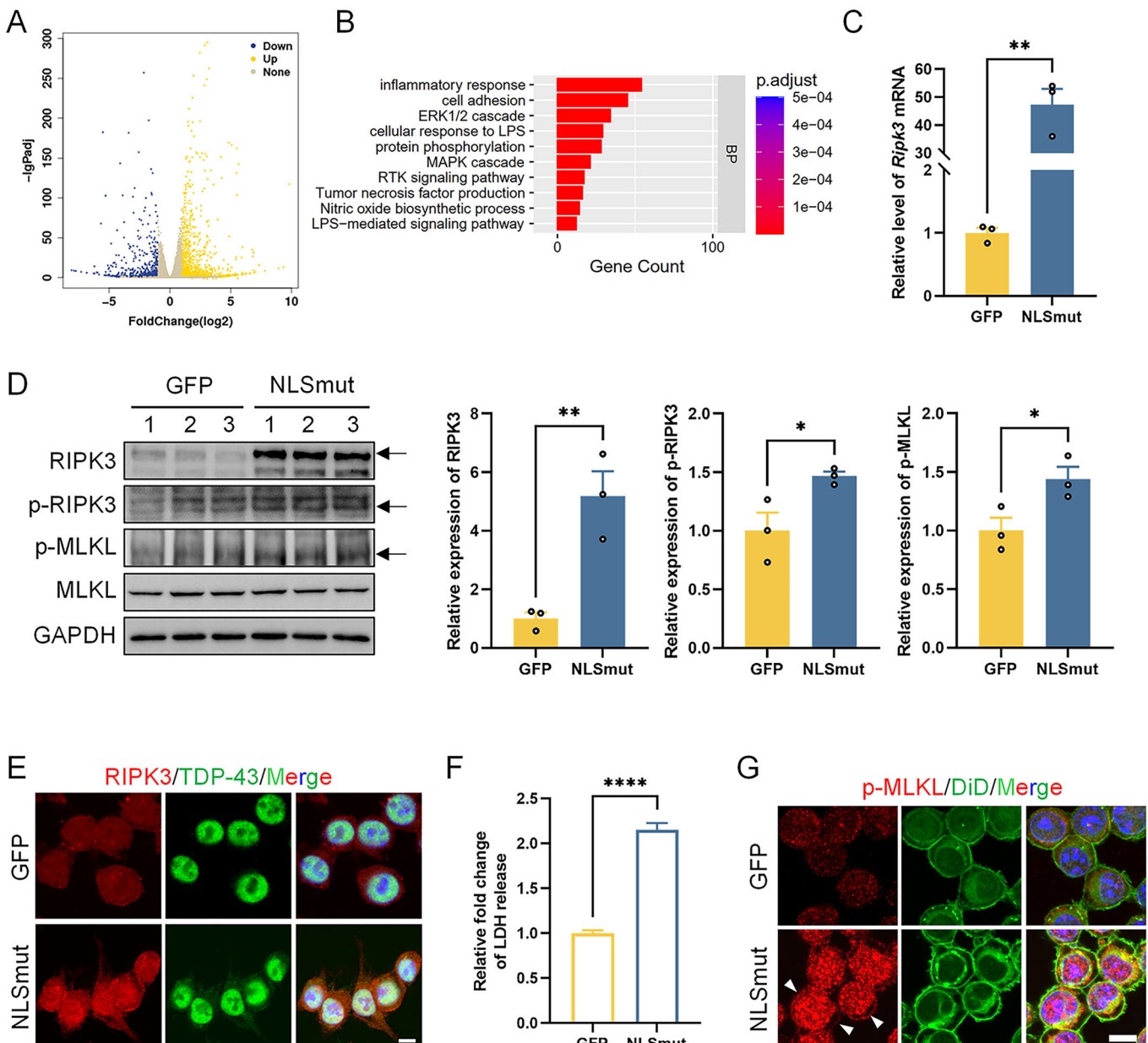

**Figure 2. TDP-43 NLSmut activated necroptosis in BV2 microglial cells.**

(A) Volcano plot visualizes differentially expressed genes (DEGs) between BV2-GFP and BV2-NLSmut cells. The yellow plots represent 909 upregulated genes, and the blue plots represent 523 downregulated genes. The effect score in the X-axis represents log2 (fold change) of gene expression. The Y-axis represents -log10 of adjusted p value. Data are from 3 biological replicates. DESeq2was used for differential gene expression analysis, p value was calculated by the Wald test, and then adjusted using the Benjamini–Hochberg (BH) method. (B) The Gene Ontology (GO) enrichment analysis of pathway preferentially enriched in BV2-NLSmut cells compared to BV2-GFP cells. The GO enrichment analysis was performed using the R package clusterProfiler, with significantly enriched results defined as those with a BH-adjusted p value < 0.05. (C) Relative mRNA level of *Ripk3* measured by qRT-PCR in BV2-GFP and BV2-NLSmut cells. Data are mean ± SEM from three biological replicates and analyzed by unpaired two-tailed Student's t test. p = 0.0012. (D) Immunoblotting using antibody recognizing RIPK3 (p = 0.0084), phospho-RIPK3 (p-RIPK3) at the Thr231/Ser232 site (p = 0.0430), phospho-MLKL (p-MLKL) at the Ser345 site (p = 0.0422), MLKL and GAPDH in BV2-GFP and BV2-NLSmut cells, along with the corresponding quantification. Data are mean ± SEM from 3 biological replicates and analyzed by unpaired two-tailed Student's t test. (E) Immunofluorescence of RIPK3 (red) and TDP-43 (green) in BV2-GFP and BV2-NLSmut cells. Hoechst (blue) for nuclei. Scale bar, 10 μm. (F) Fold change of LDH released by BV2-GFP and BV2-NLSmut cells. Data are mean ± SEM from five biological replicates and analyzed by unpaired two-tailed Student's t test. p = 0.0000006. (G) Immunofluorescence of p-MLKL (red) and DiD (green) in BV2-GFP and BV2-NLSmut cells. Hoechst (blue) for nuclei. Arrows indicate membrane-localized p-MLKL. Scale bar, 10 μm. *p < 0.05, **p < 0.01, ****p < 0.0001. Source data are available online for this figure.

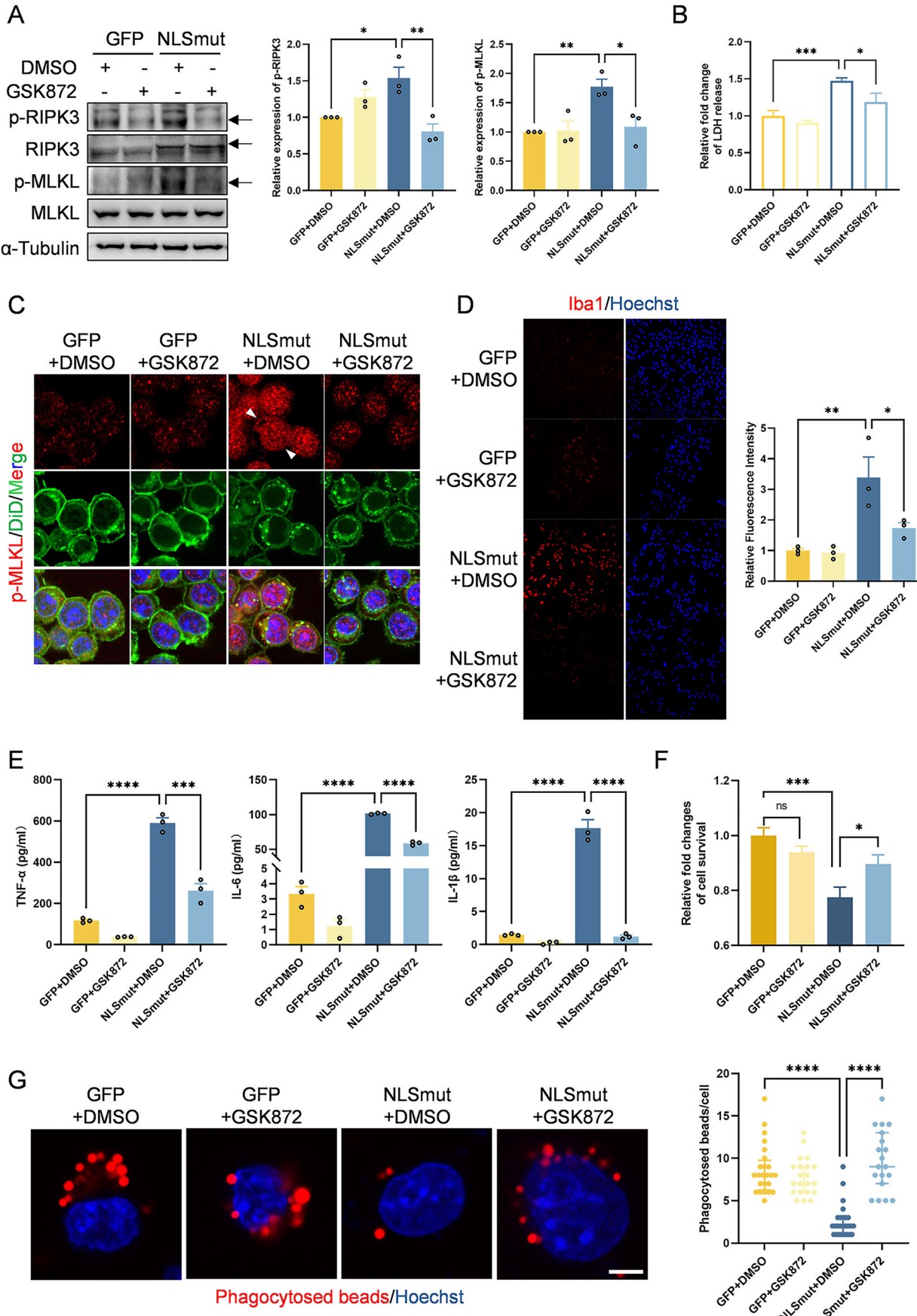

**Figure 3. Pharmacologically inhibiting necroptosis suppressed TDP-43 NLSmut induced neuroinflammation, cell death and phagocytosis inhibition.**

(A) Immunoblotting using antibody recognizing phospho-RIPK3 (p-RIPK3) at the Thr231/Ser232 site (GFP + DMSO vs. NLSmut+DMSO, $p = 0.0236$; NLSmut+DMSO vs. NLSmut+GSK872, $p = 0.0056$), RIPK3, phospho-MLKL (p-MLKL) at the Ser345 site (GFP + DMSO vs. NLSmut+DMSO, $p = 0.0082$; NLSmut+DMSO vs. NLSmut +GSK872, $p = 0.0145$), MLKL and α-Tubulin in BV2-GFP and BV2-NLSmut cells treated with or without GSK872 (5 μM, 24 h), along with the corresponding quantification. Data are mean ± SEM from 3 biological replicates and analyzed by ANOVA followed by Bonferroni's multiple comparisons. (B) Fold change of LDH released from BV2-GFP and BV2-NLSmut cells treated with or without GSK872 (5 μM, 24 h). Data are mean ± SEM from 5 biological replicates and analyzed by ANOVA followed by Bonferroni's multiple comparisons. GFP + DMSO vs. NLSmut+DMSO, $p = 0.0006$; NLSmut+DMSO vs. NLSmut+GSK872, $p = 0.0270$. (C) Immunofluorescence staining of p-MLKL (red) and DiD (green) in BV2-GFP and BV2-NLSmut cells treated with or without GSK872 (5 μM, 24 h). Hoechst (blue) for nuclei. Scale bar, 10 μm. (D) Immunofluorescence staining and the corresponding quantification of Iba1 (red) in BV2-GFP and BV2-NLSmut cells treated with or without GSK872 (5 μM, 24 h). Hoechst (blue) for nuclei. Scale bar, 100 μm. Three fields of view were analyzed for fluorescent intensities. Data are mean ± SEM and analyzed by ANOVA followed by Bonferroni's multiple comparisons. GFP + DMSO vs. NLSmut+DMSO, $p = 0.0027$; NLSmut+DMSO vs. NLSmut+GSK872, $p = 0.0210$. (E) ELISA analysis of TNF-α (GFP + DMSO vs. NLSmut+DMSO, $p = 0.00002$; NLSmut+DMSO vs. NLSmut+GSK872, $p = 0.0001$), IL-6 (GFP + DMSO vs. NLSmut+DMSO, $p = 0.000000001$; NLSmut+DMSO vs. NLSmut+GSK872, $p = 0.0000002$) and IL-1β (GFP + DMSO vs. NLSmut+DMSO, $p = 0.00001$; NLSmut+DMSO vs. NLSmut+GSK872, $p = 0.00001$) secreted by BV2-GFP and BV2-NLSmut cells treated with or without GSK872 (5 μM, 24 h). Data are mean ± SEM from three biological replicates and analyzed by ANOVA followed by Bonferroni's multiple comparisons. (F) Cell viability of BV2-GFP cells and BV2-NLSmut cells treated with or without GSK872 (5 μM, 24 h) measured by CCK-8 assay. Data are mean ± SEM from 5 biological replicates and analyzed by ANOVA followed by Bonferroni's multiple comparisons. GFP + DMSO vs. NLSmut+DMSO, $p = 0.0009$; NLSmut+DMSO vs. NLSmut+GSK872, $p = 0.0494$. (G) Immunofluorescence and the corresponding quantification indicating phagocytosed beads (red) in BV2-GFP and BV2-NLSmut cells treated with or without GSK872 (5 μM, 24 h). Hoechst (blue) for nuclei. Scale bar, 10 μm. At least 15 cells of each group were analyzed. Non-normally distributed data were presented as median and interquartile range, and analyzed by Kruskal–Wallis H test. GFP + DMSO vs. NLSmut+DMSO, $p = 0.00000004$; NLSmut+DMSO vs. NLSmut+GSK872, $p = 0.000000004$. *$p < 0.05$, **$p < 0.01$, ***$p < 0.001$, ****$p < 0.0001$; ns, not significant. Source data are available online for this figure.

the largest subunit of RNA polymerase II, at these sites were found to be significantly enhanced (Fig. 5F). We also examined the expression levels of Rpb1 and observed no significant differences between BV2-NLSmut and BV2-GFP control cells (Appendix Fig. S3), thereby excluding the influence of RNA polymerase II expression levels on *Ripk3* transcription. These findings suggest that the reduction in nuclear TDP-43 decreases its occupancy at the *Ripk3* locus, thereby facilitating the recruitment of RNA polymerase II.

Previous studies have established that (GT)n sequences in gene promoters serve as canonical binding sites for TDP-43 through its RNA recognition motifs (RRMs), with RRM1 playing a dominant role (Kuo et al, 2014). It is notable that a $(GT)_{17}$ element is located -1.0 kb upstream of the TSS within the 5′-untranslated region (UTR) of the *Ripk3* gene (Fig. 5E). To ascertain whether TDP-43 RRM1 is capable of binding to the $(GT)_{17}$ sequence of *Ripk3* promoter, we generated a GST-RRM1 domain fusion protein and evaluated its interaction with the $(GT)_{17}$ sequence in vitro (Fig. 5G). Electrophoretic mobility shift assays (EMSA) demonstrated that the RRM1 domain specifically binds to the $(GT)_{17}$ sequence within *Ripk3* DNA promoter region (Fig. 5H), thereby elucidating the molecular basis of occupancy and repressive effect of TDP-43 protein on *Ripk3* promoter.

Collectively, we propose a mechanistic cascade whereby the reduction of nuclear TDP-43, commonly accompanied by its aberrant cytoplasmic mis-localization and potentially induced by this mis-localization, results in decreased binding at the $(GT)_{17}$-containing *Ripk3* promoter via its RRM1 domain. This reduction in occupancy subsequently promotes the assembly of RNA polymerase and the initiation of transcription, ultimately leading to the upregulation of *Ripk3* transcription.

## Pharmacologically inhibiting RIPK3 kinase effectively inhibited TDP-43 NLSmut mediated neuroinflammation and cognitive impairment in mice

In order to model limbic-predominant age-related TDP-43 encephalopathy (LATE), we focused on the hippocampus, a brain region critically involved in the spatial memory deficits seen in this disease. Furthermore, to model the disease context in which TDP-43 pathology typically originates in neurons, we overexpressed TDP-43 NLSmut in neurons using a neuron-specific promoter via AAV injection, with a GFP-expressing AAV as the control.

Immunofluorescence staining revealed a significant cytoplasmic mis-localization of TDP-43 and a concomitant loss of nuclear TDP-43 expression in neurons and Iba1-positive microglia of mice injected with TDP-43 NLSmut virus (Figs. EV3 and EV4). Furthermore, we documented significant neuroinflammation, supported by the substantial increase in Iba1 immunofluorescence intensity and their ameboid shape within the hippocampus of TDP-43-NLSmut mice compared to GFP control mice (Fig. 6A). Critically, these activated microglia proximally to affected neurons exhibited significantly elevated expression levels of key necroptotic markers, including RIPK3, phosphorylated RIPK3 and phosphorylated MLKL (Fig. 6A,B).

Treatment of the RIPK3 kinase inhibitor GSK872 effectively suppressed the phosphorylated RIPK3 and MLKL in the hippocampus of TDP-43 NLSmut mice, indicating the inhibition of RIPK3-dependent necroptosis (Fig. 6B). Additionally, a significant decrease in Iba1 immunofluorescence intensity indicated that GSK872 treatment markedly inhibited neuroinflammation in TDP-43 NLSmut mice (Fig. 6A). Morris water maze (MWM) test was performed to evaluate the therapeutic efficacy of GSK872 in ameliorating spatial memory (Fig. 7A,B). No significant differences in swim speed or escape latency were observed between TDP-43 NLSmut mice and GFP controls, nor between GSK872 treatment and vehicle controls (Fig. 7C,D). However, TDP-43 NLSmut mice spent considerably less time and less distance in the target quadrant, where the platform was located, compared to GFP controls, indicating impaired spatial memory (Fig. 7B,E,F). Notably, pharmacological inhibition of RIPK3 kinase with GSK872 treatment significantly mitigated this impairment (Fig. 7E,F). These findings suggest that inhibiting necroptosis might effectively reduce neuroinflammation and cognitive deficits in mice with TDP-43 pathology.

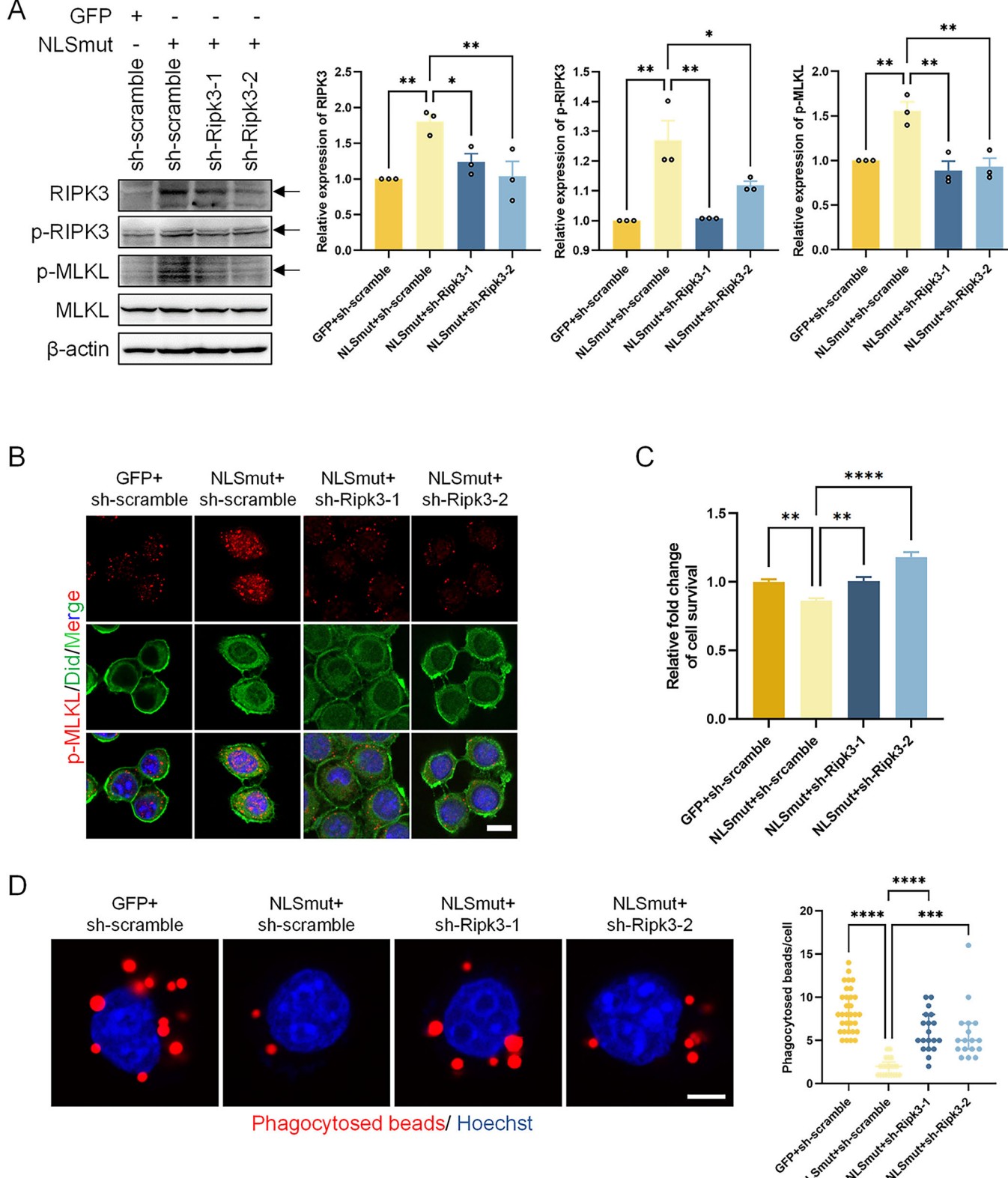

**Figure 4. Genetically inhibiting necroptosis suppressed TDP-43 NLSmut induced cell death and phagocytosis inhibition.**

(A) Immunoblotting using antibody recognizing RIPK3 (GFP+sh-scramble vs. NLSmut+sh-scramble, $p = 0.0067$; NLSmut+sh-scramble vs. NLSmut+sh-Ripk3-1, $p = 0.0440$; NLSmut+sh-scramble vs. NLSmut+sh-Ripk3-2, $p = 0.0087$), phospho-RIPK3 (p-RIPK3) at the Thr231/Ser232 site (GFP+sh-scramble vs. NLSmut+sh-scramble, $p = 0.0013$; NLSmut+sh-scramble vs. NLSmut+sh-Ripk3-1, $p = 0.0016$; NLSmut+sh-scramble vs. NLSmut+sh-Ripk3-2, $p = 0.0377$), phospho-MLKL (p-MLKL) at the Ser345 site (GFP+sh-scramble vs. NLSmut+sh-scramble, $p = 0.0051$; NLSmut+sh-scramble vs. NLSmut+sh-Ripk3-1, $p = 0.0016$; NLSmut+sh-scramble vs. NLSmut+sh-Ripk3-2, $p = 0.0025$), MLKL and β-actin in GFP+sh-scramble, NLSmut+sh-scramble, NLSmut+sh-Ripk3-1 and NLSmut+sh-Ripk3-2 cells, along with the corresponding quantification. Data are mean ± SEM from 3 biological replicates and analyzed by ANOVA followed by Bonferroni's multiple comparisons. (B) Immunofluorescence of p-MLKL (red) and DiD (green) in GFP+sh-scramble, NLSmut+sh-scramble, NLSmut+sh-Ripk3-1 and NLSmut+sh-Ripk3-2 cells. Hoechst (blue) for nuclei. Scale bar, 10 μm. (C) Cell viability of GFP+sh-scramble, NLSmut+sh-scramble, NLSmut+sh-Ripk3-1 and NLSmut+sh-Ripk3-2 cells measured by CCK-8 assay. Data are mean ± SEM from 5 biological replicates and analyzed by ANOVA followed by Bonferroni's multiple comparisons. GFP+sh-scramble vs. NLSmut+sh-scramble, $p = 0.0051$; NLSmut+sh-scramble vs. NLSmut+sh-Ripk3-1, $p = 0.0079$; NLSmut+sh-scramble vs. NLSmut+sh-Ripk3-2, $p = 0.000001$. (D) Immunofluorescence and the corresponding quantification indicating phagocytosed beads (red) in GFP+sh-scramble, NLSmut+sh-scramble, NLSmut+sh-Ripk3-1 and NLSmut+sh-Ripk3-2 cells. Hoechst (blue) for nuclei. Scale bar, 10 μm. At least 15 cells of each group were analyzed. Non-normally distributed data were presented as median and interquartile range, and analyzed by Kruskal–Wallis H test. GFP+sh-scramble vs. NLSmut+sh-scramble, $p = 0.0000000000005$; NLSmut+sh-scramble vs. NLSmut+sh-Ripk3-1, $p = 0.00004$; NLSmut+sh-scramble vs. NLSmut+sh-Ripk3-2, $p = 0.0008$. $*p < 0.05$, $**p < 0.01$, $****p < 0.0001$. sh-scramble: scramble shRNA; sh-Ripk3-1: Ripk3 shRNA-1; sh-Ripk3-2: Ripk3 shRNA-2. Source data are available online for this figure.

## Discussion

Our study demonstrated that mis-localized TDP-43 drives microglial necroptosis through RIPK3/MLKL activation, establishing a novel mechanism by which TDP-43 pathology exacerbated neuroinflammation, promoted cell death, and impaired phagocytic function in microglial cells. Importantly, pharmacologically inhibition of RIPK3 with GSK872 not only suppressed necroptotic markers but also remarkably improved microglial functions in cells exhibiting TDP-43 mis-localization. Our findings linked the nuclear depletion of TDP-43 driving *Ripk3* transcription de-repression to the activation of microglial necroptosis. The involvement of RIPK3 mediated microglial necroptosis has also been substantiated in murine models with cytoplasmic mis-localization of TDP-43. Treatment with GSK872 not only diminished necroptotic markers and neuroinflammation, but also preserved cognitive function, highlighting the therapeutic potential of targeting this pathway.

Our findings identified that necroptosis is a pivotal event in microglial cells with TDP-43 mis-localization. As a regulated form of cell death, necroptosis is characterized by activation of the RIPK1-RIPK3-MLKL signaling pathway, ultimately leading to the disruption of membrane integrity and release of pro-inflammatory cytokines (Bertheloot et al, 2021). Consistent with previous studies (Fang et al, 2021; Huang et al, 2023; Huang et al, 2018), necroptosis in microglia could lead to substantial cell death and neuroinflammation. The release of pro-inflammatory mediators, including TNF-α and IL-1β, by necroptotic microglia may establish a self-amplifying cycle of neuroinflammation that continuously exacerbates inflammatory damage. Furthermore, we observed a marked impairment in the phagocytic capacity of necroptotic BV2 microglia, suggesting a compromised ability to clear abnormal protein aggregates and cellular debris. Thereby, the heightened proinflammatory signaling coupled with diminished clearance function positions necroptotic microglia as both contributors to and victims of a deteriorating neuro-environment. This dynamic establishes a pathological feedback loop that accelerates the progression of neurodegenerative disease.

The association between necroptosis and TDP-43 pathology has been supported by previous observations: phosphorylated MLKL levels are elevated and colocalize with phosphorylated TDP-43 in TDP-43 pathology-positive neurons derived from cases of AD or ALS/FTLD (Koper et al, 2022; Van Schoor et al, 2021). Activated

RIPK1-mediated necroptosis has also been reported in mature oligodendrocytes with TDP-43 depletion (Wang et al, 2018). Studies have revealed that TDP-43 pathology could induce necroptosis through a mitochondrial-dependent manner, where elevated mitochondrial ROS leads to increased MLKL phosphorylation (Yang et al, 2024). Furthermore, consistent with evidence that TDP-43 pathology upregulates TNF-α (Herman et al, 2012; Yu et al, 2020; Zhao et al, 2015), our data show that microglia with mis-localized TDP-43 secrete high levels of TNF-α. This self-produced cytokine likely acts in an autocrine manner, binding to TNFR1 and initiating the canonical RIPK1-dependent signaling that converges on RIPK3 activation. In contrast to these indirect regulation, our study uncovers a more direct and fundamental link: TDP-43 itself acts as a transcriptional repressor of the core necroptotic executor *Ripk3*. We demonstrated that TDP-43 binds to the *Ripk3* gene locus and suppresses its transcription. Consequently, the reduction of nuclear TDP-43 derepresses *Ripk3*, leading to accumulation of RIPK3 protein and the subsequent increases in phosphorylated RIPK3 and phosphorylated MLKL.

This finding uncovers a direct mechanistic link between TDP-43 pathology and necroptosis activation, which not only advances our comprehension of their interplay that underlying the association observed in patient-derived cells but also extending our understanding of the pathologic mechanisms underlying TDP-43 pathology. Notably, TDP-43 has been recognized as a transcriptional repressor that inhibits gene transcription by occupying the gene promoters and pausing RNA polymerase (Ayala et al, 2008; Lalmansingh et al, 2011; Swain et al, 2016). Our findings corroborate this known function by identifying *Ripk3* as a direct transcriptional target. By identifying the RRM1 domain as a critical mediator of this interaction, our work provides a molecular basis for how TDP-43 dysfunction—specifically, the loss of nuclear function—propagates necroptotic signaling. It also highlights a potential target for disrupting the pathogenic crosstalk between TDP-43 loss of function and regulated necroptotic cell death in neurodegenerative conditions characterized by TDP-43 mis-localization.

Inhibition of necroptosis has proven effective across multiple neurodegenerative disease models. Targeting either RIPK1 or RIPK3 has been shown to decrease Aβ and tau pathologies, eliminate neuroinflammation and improve cognitive impairment in AD mouse models (Park et al, 2021; Yang et al, 2017). Additionally, this approach

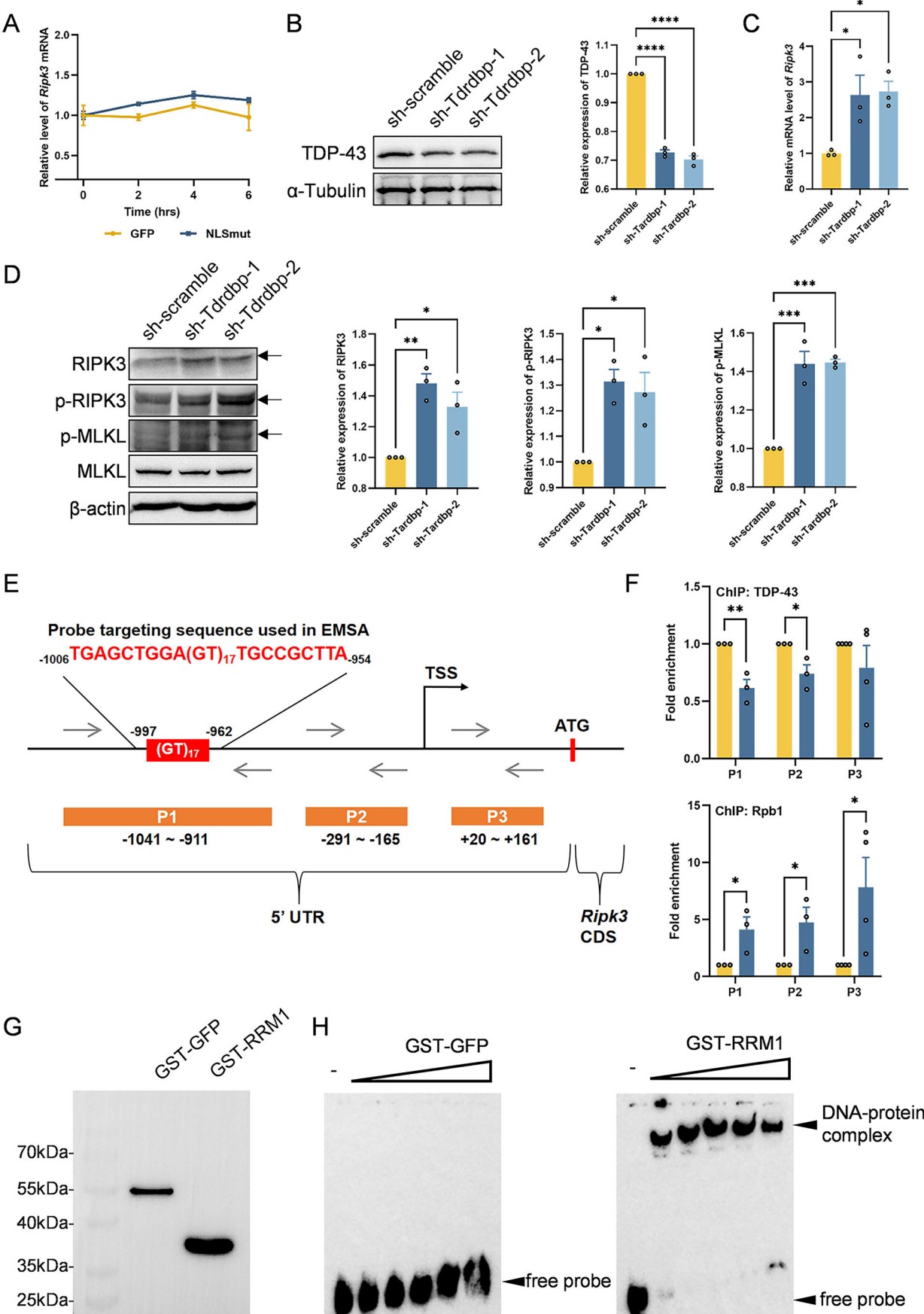

Figure 5. Decreased nuclear TDP-43 expression enhanced *Ripk3* transcription and induced necroptosis.

(A) Relative mRNA level of *Ripk3* measured by qRT-PCR in BV2-GFP and BV2-NLSmut cells treated with 5 µg/ml Actinomycin D (Act D) for 2, 4 or 6 h. Data are mean ± SEM from 3 biological replicates. (B) Immunoblotting using antibody recognizing TDP-43 and α-Tubulin in BV2 cells transfected with sh-scramble, sh-Tdrdbp-1 and sh-Tdrdbp-2, along with the corresponding quantification. Data are mean ± SEM from 3 biological replicates and analyzed by ANOVA followed by Bonferroni's multiple comparisons. sh-scramble vs. sh-Tdrdbp-1, $p = 0.000001$; sh-scramble vs. sh-Tdrdbp-2, $p = 0.0000006$. (C) Relative mRNA level of *Ripk3* measured by qRT-PCR in BV2 cells transfected with sh-scramble, sh-Tdrdbp-1 and sh-Tdrdbp-2. Data are mean ± SEM from 3 biological replicates and analyzed by ANOVA followed by Bonferroni's multiple comparisons. sh-scramble vs. sh-Tdrdbp-1, $p = 0.0367$; sh-scramble vs. sh-Tdrdbp-2, $p = 0.0293$. (D) Immunoblotting using antibody recognizing RIPK3 (sh-scramble vs. sh-Tdrdbp-1, $p = 0.0040$; sh-scramble vs. sh-Tdrdbp-2, $p = 0.0239$), phospho-RIPK3 (p-RIPK3) at the Thr231/Ser232 site (sh-scramble vs. sh-Tdrdbp-1, $p = 0.0110$; sh-scramble vs. sh-Tdrdbp-2, $p = 0.0209$), phospho-MLKL (p-MLKL) at the Ser345 site (sh-scramble vs. sh-Tdrdbp-1, $p = 0.0003$; sh-scramble vs. sh-Tdrdbp-2, $p = 0.0003$), MLKL and β-actin in BV2 cells transfected with sh-scramble, sh-Tdrdbp-1 and sh-Tdrdbp-2, along with the corresponding quantification. Data are mean ± SEM from 3 biological replicates and analyzed by ANOVA followed by Bonferroni's multiple comparisons. (E) Schematic image showing EMSA probe and fragments for ChIP-qPCR amplification in 5′-UTR of the *Ripk3* gene. (F) ChIP-qPCR analysis of TDP-43 and Rpb1 binding at the promoter regions of *Ripk3* in BV2-GFP and BV2-NLSmut cells. Data are mean ± SEM from at least 3 biological replicates and analyzed by ANOVA followed by Bonferroni's multiple comparisons. For P1 of TDP-43 binding, $p = 0.0069$; for P2 of TDP-43 binding, $p = 0.0292$; for P1 of Rpb1 binding, $p = 0.0461$; for P2 of Rpb1 binding, $p = 0.0497$; for P3 of Rpb1 binding, $p = 0.0227$. (G) Immunoblotting using antibody recognizing GST from GST-GFP or GST-RRM1 protein. (H) EMSA detecting the binding of GST-GFP or GST-RRM1 protein to the $(GT)_{17}$ sequence at the promoter regions of *Ripk3*. $*p < 0.05$, $**p < 0.01$, $***p < 0.001$, $****p < 0.0001$. sh-scramble scramble shRNA, sh-Tdrdbp-1 Tdrdbp shRNA-1, sh-Tdrdbp-2 Tdrdbp shRNA-2. Source data are available online for this figure.

has been effective in reducing dopaminergic neuronal cell death in a PD model (Iannielli et al, 2018) and in alleviating oligodendrocyte death, axonal degeneration and neuroinflammation in an ALS model (Zhu et al, 2011). Consistent with these findings, our in vitro studies using BV2 microglia showed that pharmacological inhibition of necroptosis with GSK872 significantly diminished the release of pro-inflammatory cytokine while restoring cell survival and phagocytic function. In our mouse model featuring cytoplasmic mis-localization of TDP-43, GSK872 administration led to a significant reduction in microglial necroptosis markers. Crucially, this was accompanied by attenuation of neuroinflammation and restoration of cognitive function, reaffirming the critical microglia-neuron interplay in this pathogenic context and further supporting the therapeutic potential of targeting microglial necroptosis in TDP-43-associated neurodegeneration.

A notable limitation of GSK872 is its inability to penetrate the blood-brain barrier, which necessitated the use of intracerebroventricular administration in our research. This highlights the necessity for developing new necroptosis inhibitors with improved blood-brain barrier permeability in future drug development efforts. Additionally, it is worthwhile to investigate the potential of combining necroptosis inhibition with strategies targeting other facets of the identified mechanism. For example, enhancing TDP-43 mediated transcriptional repression or facilitating TDP-43 nuclear import could be considered. Such combinatorial approaches may synergistically mitigate neurodegenerative progression by simultaneously addressing both necroptotic signaling and TDP-43 dysregulation.

# Methods

### Reagents and tools table

| Reagent/resource | Reference or source | Identifier or catalog number |
|---|---|---|
| **Experimental models** | | |
| BV2 microglial cell | Ubigene | YC-C035 |
| C57BL/6 J mice (*M. musculus*) | Shanghai Jihui Laboratory Animal Care Co., Ltd. | |

| Reagent/resource | Reference or source | Identifier or catalog number |
|---|---|---|
| **Recombinant DNA** | | |
| lentiCas9-Blast | Addgene | 52962 |
| lenti-myc-GFP | This study | |
| lenti-myc-TDP43 NLSmut | This study | |
| pCDH-EF1α-MCS-T2A-GFP | System Biosciences | CD526A-1 |
| pCDH-GST-GFP | This study | |
| pCDH-GST-RRM1 | This study | |
| **Antibodies** | | |
| Anti-myc | Merck | 05-724 |
| Anti-GFP | Santa Cruz | sc-9996 |
| Anti-TDP-43 | Proteintech | 12892-1-AP |
| Anti-Iba1 | Cell Signaling Technology | 17198 |
| Anti-p-MLKL(Ser345) | Cell Signaling Technology | 37333 |
| Fluor 488-conjugated secondary antibodies | Cell Signaling Technology | 4412 |
| Fluor 555-conjugated secondary antibodies | Cell Signaling Technology | 4413 |
| Goat anti-mouse IgG HRP-conjugated secondary antibody | Thermo Fisher Scientific | 31430 |
| Goat anti-rabbit IgG HRP-conjugated secondary antibody | Thermo Fisher Scientific | 31460 |
| Anti-p-TDP-43 | BioLegend | 829901 |
| Anti-p-RIPK3(T231/S232) | ABclonal | AP1408 |
| Anti-RIPK3 | Proteintech | 17563-1-AP |
| Anti-MLKL | ABclonal | A21894 |
| Anti-Lamin B1 | Proteintech | 66095-1-Ig |
| Anti-TNF-α | Abcam | ab66579 |
| Anti-Rpb1 CTD (4H8) | Cell Signaling Technology | 2629 |
| Anti-α-Tubulin | Cell Signaling Technology | 3873 |
| Anti-β-actin | ABclonal | AC026 |
| Anti-GAPDH | ABclonal | AC002 |

| Reagent/resource | Reference or source | Identifier or catalog number |
|---|---|---|
| Anti-NeuN | Cell Signaling Technology | 24307 |
| Anti-FLAG | Sigma | F1804 |
| **Oligonucleotides and other sequence-based reagents** | | |
| RIPK3 shRNA | Genomeditech | Materials and Methods-Plasmids |
| TDP-43 shRNA | Genomeditech | Materials and Methods-Plasmids |
| qRT-PCR primers | This study | Table EV1 |
| ChIP-qPCR primers | This study | Table EV2 |
| EMSA probe | This study | Materials and Methods-EMSA |
| **Chemicals, enzymes and other reagents** | | |
| Hoechst 33258 | Yeasen | 40730ES03 |
| GSK872 | Selleck | S8465 |
| Trizol | Vazyme | R411-01 |
| Latex beads, fluorescent red | Sigma | L2778 |
| protein A/G agarose beads | Santa Cruz | sc-2003 |
| Glutathione Sepharose | Cytiva | 17075601 |
| AAV2/9-hSyn-EGFP-P2A-3×FLAG | OBiO Technology | |
| AAV2/9-hSyn-EGFP-P2A-TDP43-NLS-3×FLAG | OBiO Technology | |
| **Software** | | |
| SuperMaze | Shanghai Xinruan Information Technology Co., Ltd | |
| GraphPad Prism 9 | https://www.graphpad.com/features | |
| ImageJ | https://imagej.net/ij/ | |
| SPSS 26 | https://www.ibm.com/cn-zh/products/spss-statistics | |
| **Other** | | |
| Cell Plasma Membrane Staining Kit with DiD | Beyotime | C1995S |
| Cell Counting Kit-8 | Beyotime | C0038 |
| TNF-α ELISA Kit | ABclonal | RK00027 |
| IL-6 ELISA Kit | ABclonal | RK00008 |
| IL-1β ELISA Kit | ABclonal | RK00006 |
| QuickChange Site-Directed Mutagenesis Kit | Agilent | 200521 |
| PrimeScript™ RT reagent Kit | Takara | RR047A |
| TB Green® Premix Ex Taq™ II | Takara | RR420A |
| SimpleChIP® Enzymatic Chromatin IP Kit | Cell Signaling Technology | 9003S |
| EMSA Probe Biotin Labeling Kit | Beyotime | GS008 |
| Chemiluminescent EMSA Kit | Beyotime | GS009 |

## Plasmids

To generate the lentiviral vector expressing human TDP-43 and GFP, human TDP-43 with an N-terminal myc tag or GFP cDNA with an N-terminal myc tag was amplified using polymerase chain reaction (PCR) and subsequently cloned into the lentiviral vector via XbaI and BamHI sites. The NLS (K95A/K97A/R98A) mutations (Winton et al, 2008) were introduced by QuickChange mutagenesis. The constructs for RIPK3 shRNA-1 (targeted sequence: TTGGACCCAGAGCTGTTATTT) and shRNA-2 (targeted sequence: GTCTCCGAGGTAAAGCATTAT), as well as TDP-43 shRNA-1 (targeted sequence: GCTTTGTTCGATTTACAGAAT) and shRNA-2 (targeted sequence: GAATATGAAACCCAAGT-GAAA), were produced from Genomeditech (Shanghai, China). Additionally, GST-tagged GFP and GST-tagged RRM1 cDNA were amplified by PCR and cloned into the pCDH-EF1α-MCS-T2A-GFP vector at EcoRI sites.

## Cell culture and transfection

BV2 microglial cells were cultured in Dulbecco's Modified Eagle Medium (DMEM) supplemented with 10% fetal bovine serum (FBS) and 100 U/ml penicillin and 100 mg/ml streptomycin. All cells were cultured at 37 °C in a humidified incubator with 5% $CO_2$.

The second generation of lentiviral plasmids (pMD2.G and psPAX2) were utilized in conjunction with HEK293T cells. To establish stable cell lines, BV2 cells were subjected to lentiviral infection, followed by selection with 2 μg/mL puromycin.

GSK872 were supplemented to BV2 cells at a concentration of 5 μM, 24 h prior to cell harvest, with an equivalent volume of dimethyl sulfoxide (DMSO) serving as the control.

## Subcellular protein fractionation

Cells were washed with phosphate-buffered saline (PBS) and subsequently collected in a buffer containing 1.5 mM EGTA, 1 mM EDTA, 1 mM DTT, protease and phosphatase inhibitor, followed by centrifugation at 1500 rpm for 5 min at 4 °C. The pellet was resuspended by pipetting and vortexing in a buffer containing 40 mM HEPES (pH 7.5), 0.1% CA630, 5 mM EGTA, 3 mM $MgCl_2$, 10 mM NaCl, 1 mM DTT, protease and phosphatase inhibitors, and then centrifuged at $14{,}000 \times g$ for 5 min at 4 °C to isolate the cytoplasmic fraction from the supernatant. For the extraction of nuclear components, the remaining pellet was resuspended in a buffer containing 25% glycerol, 1.5 mM $MgCl_2$, 420 mM NaCl, 0.2 mM EDTA, 1 mM DTT, protease and phosphatase inhibitors, then incubated on ice for 10 min, briefly sonicated on ice, and centrifuged at $14{,}000 \times g$ for 10 min at 4 °C. The resulting supernatant was collected as the nuclear fraction.

## Subcellular RNA fractionation

Cells were harvested with 5 mM EDTA, and subjected to centrifugation at 1500 rpm for 5 min at 4 °C. The pellet was gently resuspended in gentle lysis buffer composed of 10 mM HEPES-KOH (pH 7.4), 10 mM NaCl, 3 mM $MgCl_2$ and 0.2% NP-40, followed by centrifugation at 5000 rpm for 5 min at 4 °C. The supernatant, representing the cytoplasmic fraction, was transferred to a new tube. The remaining nuclear pellet was washed once with

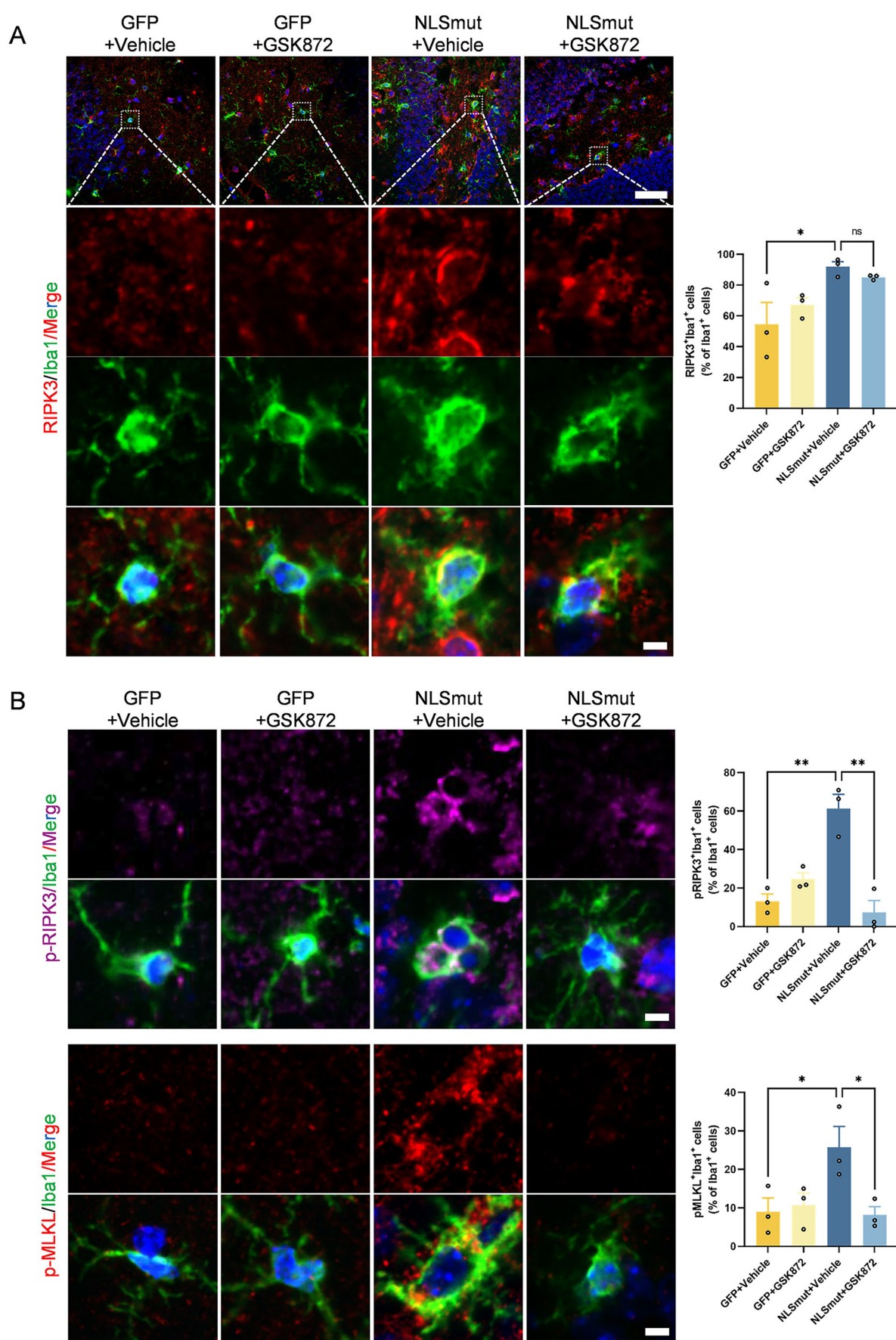

**Figure 6. GSK872 inhibited TDP-43 NLSmut mediated neuroinflammation and RIPK3-dependent necroptosis in mice.**

(A) Immunofluorescence and the corresponding quantification of RIPK3 (red) and Iba1 (green) in GFP control and TDP-43-NLSmut mice treated with or without GSK872. Hoechst (blue) for nuclei. Scale bar, 50 μm in low magnification, 5 μm in high magnification. GFP+Vehicle vs. NLSmut+Vehicle, $p = 0.0398$. (B) Immunofluorescence and the corresponding quantification of phospho-RIPK3 (p-RIPK3) (magenta) (GFP+Vehicle vs. NLSmut+Vehicle, $p = 0.0026$; NLSmut+Vehicle vs. NLSmut+GSK872, $p = 0.0014$), phospho-MLKL (p-MLKL) (red) (GFP+Vehicle vs. NLSmut+Vehicle, $p = 0.0465$; NLSmut+Vehicle vs. NLSmut+GSK872, $p = 0.0384$) and Iba1 (green) in GFP control and TDP-43-NLSmut mice treated with or without GSK872. Hoechst (blue) for nuclei. Scale bar, 5 μm. For each group, three mice were analyzed, with 3–6 fields of view assessed per mouse. Data are mean ± SEM and analyzed by ANOVA followed by Bonferroni's multiple comparisons. *$p < 0.05$, **$p < 0.01$; ns, not significant. Source data are available online for this figure.

the gentle lysis buffer and centrifuged again. Trizol reagent was added directly to both the cytoplasmic and nuclear fractions to facilitate subsequent RNA extraction.

## Immunofluorescence of cultured cells

Cells were cultured on coverslips and fixed using 4% paraformaldehyde (PFA) for 15 min. Following fixation, the cells were permeabilized in 0.2% Triton X-100 for 5 min, blocked in 1% bovine serum albumin (BSA) and 2% goat serum for 30 min, and incubated with primary antibody overnight at 4 °C. The primary antibodies include anti-myc (1:100), anti-TDP-43 (1:100), anti-Iba1 (1:100) and anti-RIPK3 (1:100).

Cell plasma membrane staining with DiD was performed in accordance with the manufacturer's instructions provided in the Cell Plasma Membrane Staining Kit with DiD. Briefly, cells were washed with PBS twice, and incubated with DiD (400×) buffer at 37°C for 15 min. Following this incubation, the cells were washed twice with PBS, fixed using iced-cold methanol at 4 °C for 15 min, and then blocked in the blocking buffer with 1% BSA and 2% goat serum for 30 min. The cells were then incubated overnight at 4 °C with primary antibodies, specially anti-p-MLKL(Ser345) (1:100).

After being washed with PBS, cells were incubated with Alexa Fluor 488- or 555-conjugated secondary antibodies (1:500). The nucleus was stained with Hoechst 33258. Cells were imaged with confocal microscopy. Images were analyzed with ImageJ software.

## Cell viability assay

Cell viability was examined by Cell Counting Kit-8 in accordance with the manufacturer's instructions. Briefly, cells were seeded in 96-well plates at a density of 5000 cells per well and subjected to GSK872 or DMSO treatment. Before cell harvest, the culture medium was replaced with 100 μL fresh DMEM supplemented with 10 μL CCK-8 reagent per well, and the plates were incubated at 37 °C for 2 h. The absorbance at 450 nm was measured using a microplate reader (BioTek, USA) to determine cell viability.

## Western blot

Samples were homogenized in protein lysis buffer composed of 125 mM Tris HCl (pH 6.8), 4% SDS, 20% glycerol, 10% 2-mercapto-ethanol, 0.02% bromophenol blue. Equal amounts of total proteins were subjected to sodium dodecyl sulfate–polyacrylamide gel electrophoresis and transferred to polyvinylidene fluoride (PVDF) membranes. After blocking, the membranes were incubated with primary antibodies overnight at 4 °C. Subsequently, the membranes were incubated with goat anti-

mouse or anti-rabbit IgG HRP-conjugated secondary antibody (1:5000) for 1 h at room temperature (RT). Proteins were visualized using chemiluminescent detection reagents. The primary antibodies include anti-myc (1:1000), anti-TDP-43 (1:1000), anti-p-TDP-43 (1:1000), anti-p-RIPK3(T231/S232) (1:1000), anti-RIPK3 (1:1000), anti-p-MLKL(Ser345) (1:1000), anti-MLKL (1:1000), anti-Lamin B1 (1:1000), anti-TNF-α (1:1000), anti-Rpb1 CTD (4H8) (1:1000), anti-α-Tubulin (1:1000), anti-β-actin (1:1000), and anti-GAPDH (1:1000).

## Enzyme-linked immunosorbent assay (ELISA)

The supernatant of cell culture medium was collected and centrifuged at 1000 g for 10 min at 4 °C. The expression levels of pro-inflammatory cytokines TNF-α, IL-6, IL-1β secreted by BV2 cells were investigated by ELISA Kit according to the manufacturer's instructions. The absorbance at 450 nm was measured using a microplate reader (BioTek, USA).

## Phagocytosis assay

Cells were seeded on coverslips in 24-well plates and subjected to GSK872 or DMSO treatment. The fluorescent latex beads are pre-opsonized in 50% FBS and PBS. Culture medium was then replaced with 200 μL DMEM alone and the pre-opsonized beads were loaded to the cells and incubated at 37 °C for 3 h. After incubation, the remaining beads are gently washed off the cells and fixed by 4% PFA at RT for 15 min. The nucleus was stained with Hoechst 33258. Cells were imaged with confocal microscopy.

## RNA extraction, cDNA preparation and quantitative real-time PCR

Total RNA was extracted using Trizol reagent. First-strand cDNA synthesis was performed using PrimeScript™ RT reagent Kit. Quantitative real-time PCR (qRT-PCR) was carried out with TB Green® Premix Ex Taq™ II on the CFX96 real-time PCR detection system (Bio-Rad). All reactions were conducted in accordance with the manufacturers' protocols. The relative mRNA expression values were normalized to *Actb* mRNA. The primer sequences used are listed in Table EV1.

## RNA sequencing

Total RNA was extracted using Trizol reagent. The transcriptome sequencing and analysis were conducted by Annoroad Gene Technology (Beijing, China). Differential expression analysis between two groups were performed using the DESeq2 software.

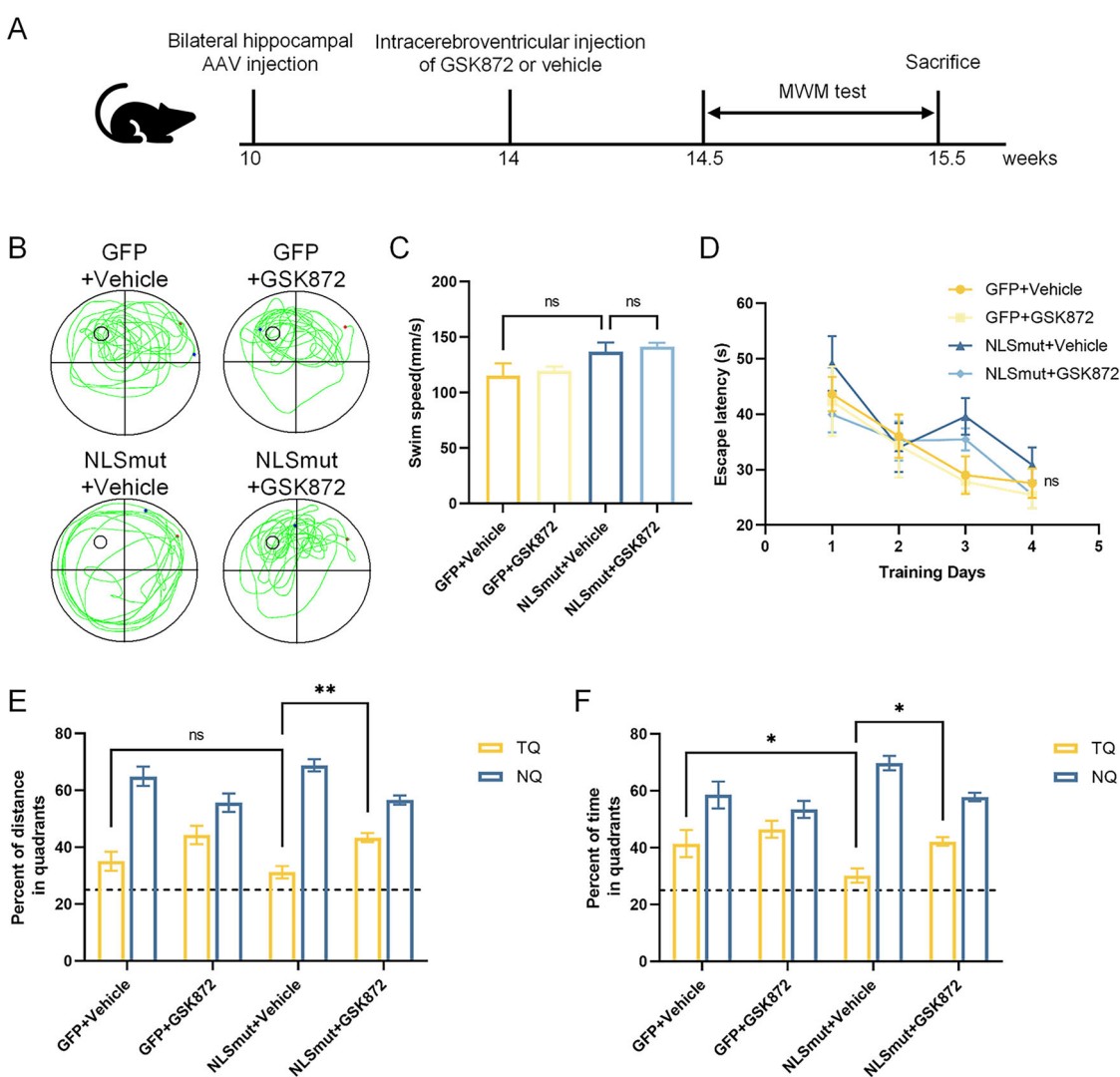

**Figure 7. GSK872 mitigated TDP-43 NLSmut induced cognitive impairment in mice.**

(A) Schematic image of experimental design. Representative paths (B), swim speed (C), escape latency (D), percent of distance in quadrants (E) and percent of time in quadrants (F) in the Morris water maze test of GFP control and TDP-43-NLSmut mice treated with or without GSK872. $n = 6$ mice for each group. Data are mean ± SEM and analyzed by ANOVA followed by Bonferroni's multiple comparisons. For percent of distance in quadrants (E), NLSmut+Vehicle vs. NLSmut+GSK872, $p = 0.0084$. For percent of time in quadrants (F), GFP+Vehicle vs. NLSmut+Vehicle, $p = 0.0418$; NLSmut+Vehicle vs. NLSmut+GSK872, $p = 0.0292$. TQ target quadrant (where the platform was located), NQ nontarget quadrant. *$p < 0.05$, **$p < 0.01$; ns, not significant. Source data are available online for this figure.

The method of Benjamini–Hochberg (BH) was employed to adjust the resulting $p$ values to control for false discovery rates. Genes with adjusted $p$ values < 0.05 and fold change > 2 ($|\text{Log2FC}| > 1$) were found to be assigned as differentially expressed. The Gene Ontology (GO) enrichment analysis of differentially expressed genes was performed using the R package clusterProfiler, with significantly enriched results defined as those with a BH-adjusted $p$ value < 0.05.

## Chromatin immunoprecipitation assay (ChIP)-qPCR

ChIP-qPCR experiments were performed using SimpleChIP® Enzymatic Chromatin IP Kit strictly followed the manufacturer's protocol. Briefly, cells were crosslinked with 1% formaldehyde for 15 min in 10 cm culture dishes at RT. Formaldehyde was stopped

by the addition of glycine, and then washed with ice-cold PBS twice. Cells were collected in ice-cold PBS, followed by centrifugation at 2000× $g$ for 5 min at 4 °C. Cells were resuspended using buffer A and incubated on ice for 10 min. Pellet nuclei were collected and resuspended with buffer B. Micrococcal nuclease was added and incubated for 20 min at 37 °C with frequent mixing to digest DNA to length of ~150–900 bp. The digestion was stopped by adding EDTA. After centrifugation at 16,000× $g$ for 1 min at 4 °C, nuclear pellet was resuspended in 1× ChIP Buffer, incubated on ice for 10 min, and sonicated. The sonicated samples were then centrifuged at 9400× $g$ for 10 min at 4 °C, and the supernatant was transferred to a new tube. The chromatin was pre-cleaned with protein A/G agarose beads. 2% of the extract was saved as the input sample. The following antibodies were added to the ChIP sample

and immunoprecipitated overnight: anti-TDP-43 and anti-Rpb1 CTD (4H8). Magnetic beads were added to each ChIP sample and rotated at 4 °C for 2 h. After washing four times, 150 µL ChIP Elution Buffer was added and vortexed gently at 1200 rpm for 30 min at 65 °C. Then proteinase K was added to reverse the crosslink at 65 °C for 2 h. DNA was purified using PCR Purification Kit. qRT-PCR was carried out using TB Green® Premix Ex Taq™ II on the CFX96 real-time PCR detection system (Bio-Rad). The primer sequences used for CHIP-qPCR are listed in Table EV2.

## Electrophoretic gel mobility shift assay (EMSA)

GST-GFP and GST-RRM1 were produced in HEK293T cells. Cell lysates were pre-cleaned with protein A/G agarose beads for 90 min at 4 °C. Glutathione Sepharose was used to pull down GST-GFP and GST-RRM1 overnight at 4 °C. GST-GFP and GST-RRM1 were then released by reduced glutathione (50 mM) in a buffer containing 200 mM NaCl, 50 mM Tris (PH 9.0) and protease inhibitors.

Single stranded DNA probe (TGAGCTGGA(GT)$_{17}$TGCCGCTTA) was labelled with biotin using EMSA Probe Biotin Labeling Kit according to the manufacturer's instructions.

EMSA was conducted using Chemiluminescent EMSA Kit in accordance with the manufacturer's instructions. Briefly, GST-GFP and GST-RRM1 protein was incubated with biotin-labeled probe in the binding buffer for 40 min at RT. The resulting reaction mixtures were subjected to 6% polyacrylamide gel and transferred to nylon membrane. Following crosslinking with UV-light, the membrane was blocked and incubated with Streptavidin-HRP Conjugate. The labeled probes were visualized using chemiluminescent detection reagents.

## Animals

Ten-week-old male C57BL/6 J mice were used in this study. The mice were maintained under controlled temperature and humidity (23 ± 2 °C, 60-70%, 12 h light/dark cycle) with free access to food and water. All animal experimentation protocols were approved by the Animal Ethical and Welfare Committee of XinHua Hospital Affiliated to Shanghai Jiao Tong University School of Medicine (XHEC-F-2024-013). A minimum of six mice were included in each treatment group for examination.

## Stereotaxic injection

The mice were randomly divided into four groups: GFP+Vehicle, GFP + GSK872, NLSmut+Vehicle, and NLSmut+GSK872. The mice were anesthetized by intraperitoneal injection of 2.5% Avertin (150 µL/ 10 g) and placed in a stereotaxic frame. In total, 1 µL of AAV2/9-hSyn-EGFP-P2A-3×FLAG or AAV2/9-hSyn-EGFP-P2A-TDP43-NLS-3×FLAG was injected bilaterally into the hippocampus, and the coordinates were as follows: anteroposterior (AP) −1.8 mm, mediolateral (ML) ± 1.8 mm, and dorsoventral (DV) −1.9 mm. GSK872 was diluted with 1% DMSO to a concentration of 25 mM. In all, 4 µL of prepared GSK872 or vehicle was injected into the lateral ventricle 4 weeks after AAV injection, and the coordinates were as follows: anteroposterior (AP) -0.1 mm, mediolateral (ML) 1 mm, and dorsoventral (DV) −2.7 mm. Injection was performed through a Hamilton 10 µL syringe with a speed of 0.2 µL/min, followed by 2 min for diffusion before the syringe was removed.

### The paper explained

**Problem**

TAR DNA-binding protein-43 (TDP-43) pathology, characterized by nuclear depletion and cytoplasmic mis-localization, is a central feature of many neurodegenerative diseases, including amyotrophic lateral sclerosis (ALS), frontotemporal dementia (FTD) and Alzheimer's disease (AD). While its role in neurons has been widely studied, how it functions in the brain's immune cells—microglia—remains poorly understood. Since microglia-driven neuroinflammation is crucial in disease progression, understanding how TDP-43 pathology disrupts these cells is essential for developing novel therapeutic strategies.

**Results**

This study uncovers a direct and novel link between TDP-43 pathology and a specific, highly inflammatory form of cell death called necroptosis in microglia. We found that the loss of TDP-43 from the nucleus drives *Ripk3* transcription derepression. The resulting overproduction of RIPK3 protein primes microglia for necroptosis, which in turn exacerbates neuroinflammation, promotes cell death, and impairs phagocytic function in microglial cells.

More importantly, we showed that blocking this pathway—either genetically or with a molecule called GSK872 that inhibits RIPK3—could restore healthy microglial function in cells. In a mouse model mimicking TDP-43 pathology, GSK872 remarkably reduced brain inflammation and rescued memory deficits.

**Impact**

Our work identifies the RIPK3-necroptosis pathway in microglia as a crucial mechanism linking TDP-43 pathology to neuroinflammation and cognitive decline. Targeting necroptosis offers a promising avenue for much-needed disease-modifying therapies.

## Morris water maze test (MWM)

The MWM test was performed in a circular pool (120 cm in diameter, 60 cm deep) filled with water kept at 21 ± 1 °C. Water was transparent on the first day (Day 0) and made opaque in the remaining test days. The MWM test contains two parts: spatial acquisition and probe trial. In spatial acquisition, mice were trained to locate a submerged platform (10 cm in diameter, positioned 1 cm below the water surface) within a 60-s cutoff period. If a mouse failed to find the platform, it was gently guided to the platform. Once on the platform, animals were given a rest for 20 s to reinforce spatial memory. Each mouse was given four trials per day for 5 consecutive days, and the trials started at random from four possible starting positions. In probe trial, the platform was removed. Each mouse was allowed to explore the pool for 120 s. Swim speed, escape latency, and distance and time in each quadrant were measured by a video-imaging system.

## Immunofluorescence of brain sections

Mice were anesthetized and perfused with precooled PBS, and the brains were postfixed overnight in 4% PFA. Sample were washed with PBS, and then processed in a sucrose gradient up to 30% for cryoprotective embedding. Sagittal sections were prepared by Cryostat (Leica). Brain sections were incubated with 3% $H_2O_2$ for 25 min to inhibit endogenous peroxidase, then blocked with 3% BSA with 0.3% Triton X-100 and 1% goat serum for 60 min and incubated with primary antibodies overnight at 4 °C. Subsequently,

the sections were incubated with HRP-linked secondary antibodies (1:100) and fluorescein tyramines. The nucleus was stained with Hoechst 33258. Sections were visualized using confocal microscopy. To better resolve the subcellular localization of TDP-43, cells co-stained for TDP-43 and Iba1 were imaged using Z-stack acquisition. Images were analyzed with ImageJ software. The primary antibodies include anti-Iba1 (1:100), anti-TDP-43 (1:100), anti-NeuN (1:100), anti-FLAG (1:100), anti-p-RIPK3(T231/S232) (1:100), anti-RIPK3 (1:100), and anti-p-MLKL(Ser345) (1:100).

## Statistical analyses

All experiments were conducted with a minimum of three biological replicates. Statistical analysis was performed using IBM SPSS v26.0 and GraphPad Prism 9. Normally distributed data were presented as mean ± SEM, and analyzed by unpaired two-tailed Student's $t$ test or one-way analysis of variance (ANOVA) followed by Bonferroni's multiple comparisons. Non-normally distributed data were presented as median and interquartile range, and analyzed by Kruskal–Wallis H test. $p < 0.05$ was considered as statistically significant (*$p < 0.05$, **$p < 0.01$, ***$p < 0.001$, ****$p < 0.0001$; ns, not significant).

## Data availability

The datasets produced in this study are available in the following databases: RNA-Seq data: Gene Expression Omnibus GSE315250.

The source data of this paper are collected in the following database record: biostudies:S-SCDT-10_1038-S44321-026-00394-9.

## Peer review information

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

## Acknowledgements

This work was supported by the National Natural Science Foundation of China (No. 82472014, 81901162), and Natural Science Foundation of Shanghai (No. 24ZR1450000, 23ZR1441200). The authors thank Dr. Xiting Pu from Renji Hospital Affiliated to Shanghai Jiao Tong University School of Medicine for her assistance in methodology. The authors thank Dr. Chonghuai Yan from Xinhua Hospital Affiliated to Shanghai Jiao Tong University School of Medicine, and Dr. Xijin Wang from Tongji Hospital Affiliated to Tongji University School of Medicine for the use of their equipment.

## Author contributions

**Shenrui Guo**: Resources; Investigation; Writing—original draft; Statistics. **Hongfu Jin**: Resources; Investigation; Statistic. **Hui Sun**: Investigation; Statistics. **Shuo Huang**: Methodology; Statistics. **Yuanyuan Chen**: Methodology; Suggestion. **Yuge Chang**: Methodology; Suggestion. **Yu Zhang**: Resources; Suggestion. **Lin Ding**: Resources; Suggestion. **Suyun Chen**: Resources; Suggestion. **Chenglai Fu**: Conceptualization; Supervision. **Yafu Yin**: Conceptualization; Supervision; Funding acquisition. **Weiwei Cheng**: Supervision; Funding acquisition; Writing—review and editing; Project administration.

Source data underlying figure panels in this paper may have individual authorship assigned. Where available, figure panel/source data authorship is listed in the following database record: biostudies:S-SCDT-10_1038-S44321-026-00394-9.

## Disclosure and competing interests statement

The authors declare no competing interests.

# Expanded View Figures

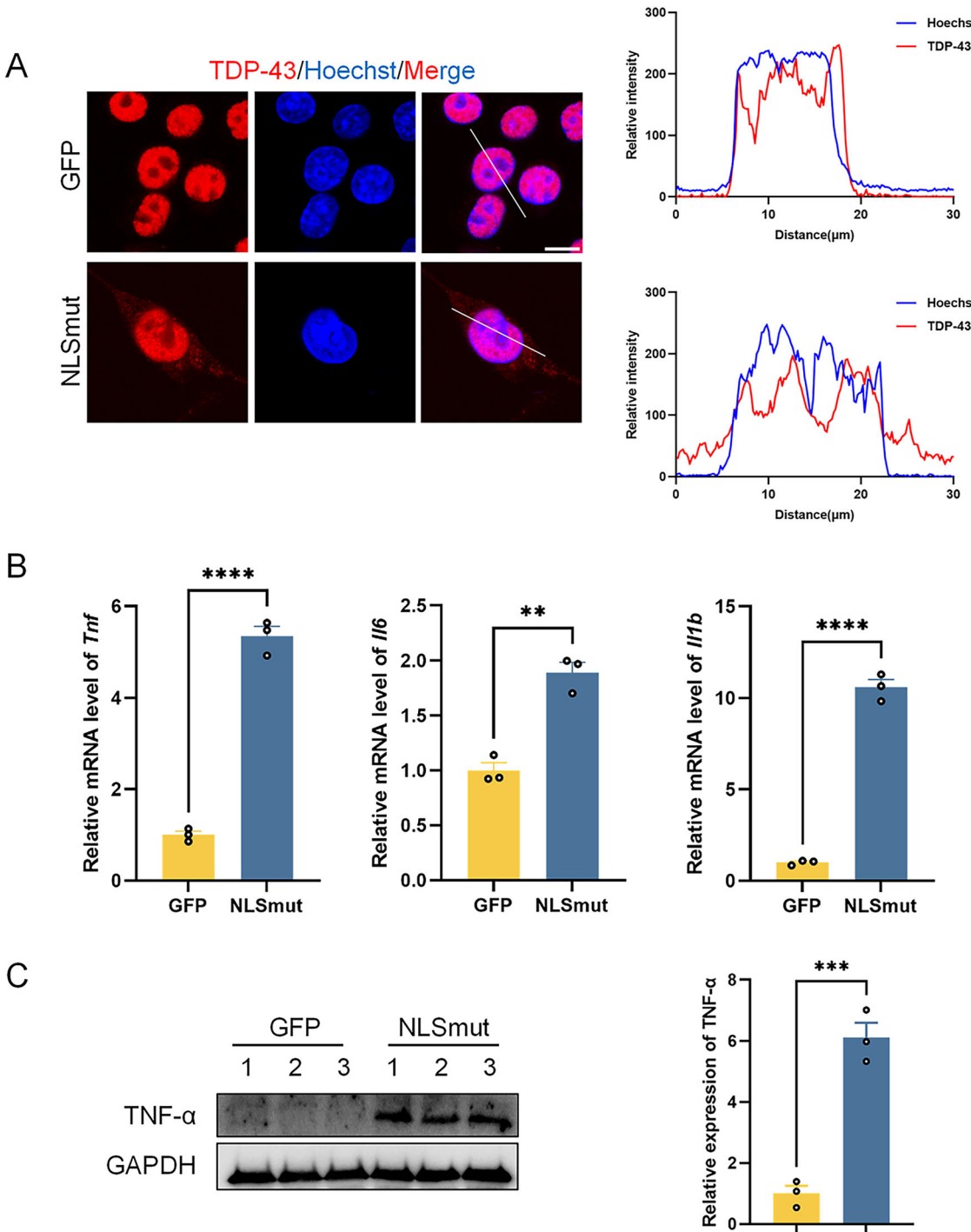

**Figure EV1.   (relative to Fig. 1) TDP-43 NLSmut induced neuroinflammation in BV2 microglial cells.**

(**A**) Immunofluorescence and colocation analysis of TDP-43 (red) and Hoechst (blue) in BV2-GFP and BV2-NLSmut cells. Scale bar, 10 μm. (**B**) Relative mRNA level of *Tnf* ($p = 0.00005$), *Il6* ($p = 0.0016$) and *Il1b* ($p = 0.00002$) measured by qRT-PCR in BV2-GFP and BV2-NLSmut cells. Data are mean ± SEM from three biological replicates and analyzed by unpaired two-tailed Student's *t* test. (**C**) Immunoblotting using antibody recognizing TNF-α and GAPDH in BV2-GFP and BV2-NLSmut cells, along with the corresponding quantification. $p = 0.0008$. Data are mean ± SEM from 3 biological replicates and analyzed by unpaired two-tailed Student's *t* test. **$p < 0.01$, ***$p < 0.001$, ****$p < 0.0001$.

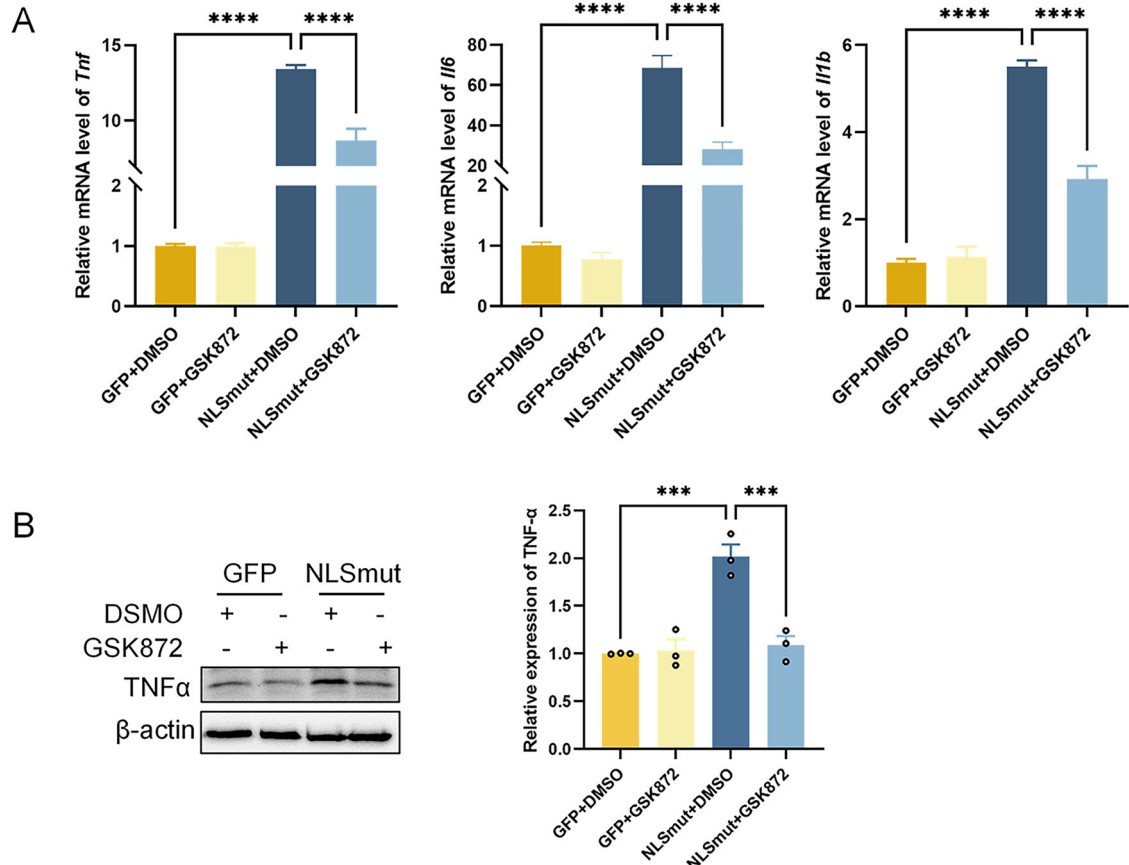

**Figure EV2.   (relative to Fig. 3) Pharmacologically inhibiting necroptosis suppressed TDP-43 NLSmut induced neuroinflammation.**

(A) Relative mRNA level of *Tnf* (GFP + DMSO vs. NLSmut+DMSO, *p* = 0.00000005; NLSmut+DMSO vs. NLSmut+GSK872, *p* = 0.00008), *Il6* (GFP + DMSO vs. NLSmut +DMSO, *p* = 0.000002; NLSmut+DMSO vs. NLSmut+GSK872, *p* = 0.00008) and *Il1b* (GFP + DMSO vs. NLSmut+DMSO, *p* = 0.0000006; NLSmut+DMSO vs. NLSmut +GSK872, *p* = 0.00004) measured by qRT-PCR in BV2-GFP and BV2-NLSmut cells treated with or without GSK872 (5 μM, 24 h). Data are mean ± SEM from three biological replicates and analyzed by ANOVA followed by Bonferroni's multiple comparisons. (B) Immunoblotting using antibody recognizing TNF-α and β-actin in BV2-GFP and BV2-NLSmut cells treated with or without GSK872 (5 μM, 24 h), along with the corresponding quantification. GFP + DMSO vs. NLSmut+DMSO, *p* = 0.0004; NLSmut+DMSO vs. NLSmut+GSK872, *p* = 0.0007. Data are mean ± SEM from 3 biological replicates and analyzed by ANOVA followed by Bonferroni's multiple comparisons. ****p* < 0.001, *****p* < 0.0001.

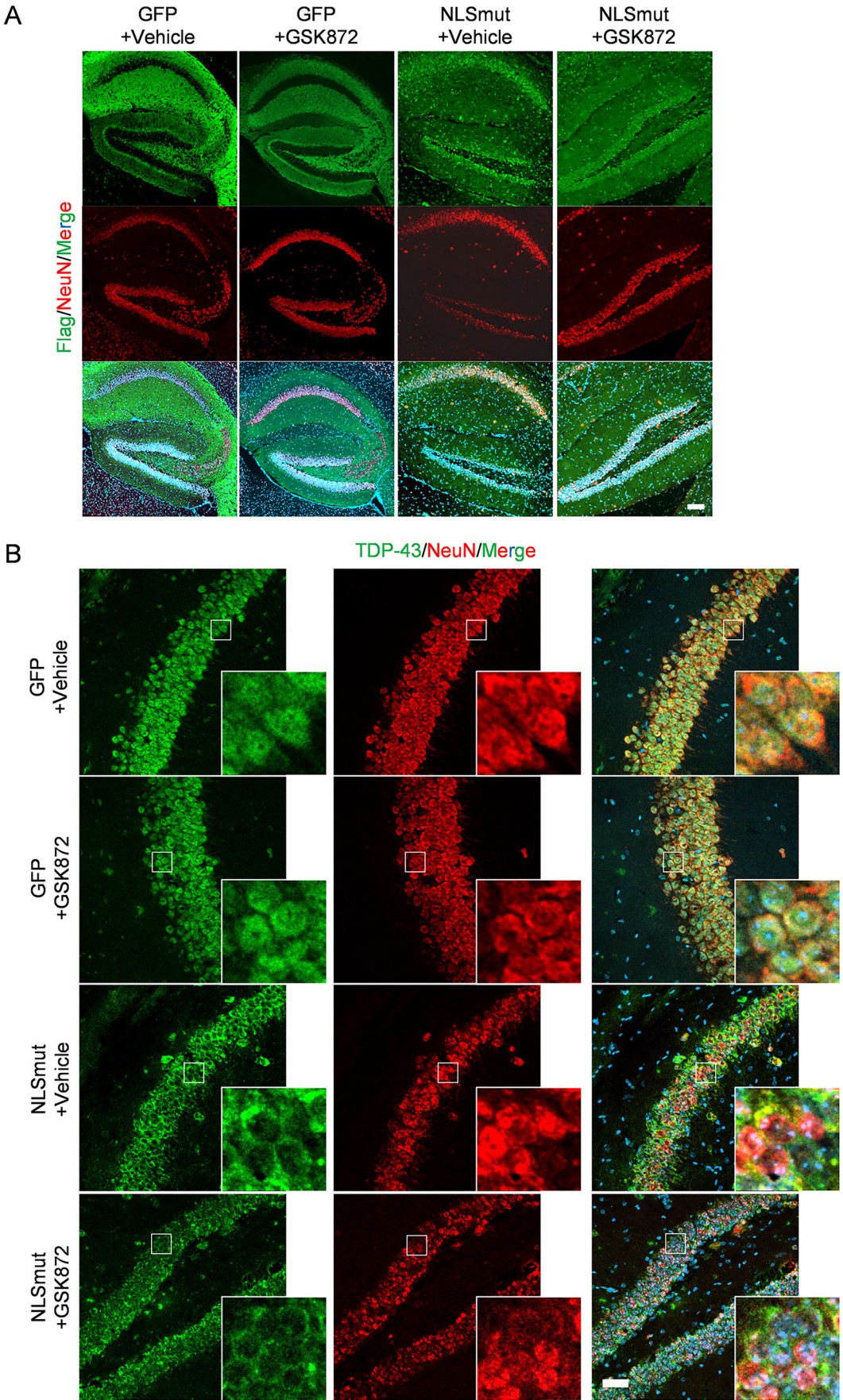

◄ **Figure EV3.** **(relative to Figs. 6 and 7) Immunofluorescence staining validated the overexpression of TDP-43 NLSmut in neurons in mice.**

(A) Immunofluorescence of Flag (green) and NeuN (red) in GFP control and TDP-43-NLSmut mice treated with or without GSK872. Hoechst (blue) for nuclei. Scale bar, 100 μm. (B) Immunofluorescence of TDP-43 (green) and NeuN (red) in GFP control and TDP-43-NLSmut mice treated with or without GSK872. Hoechst (blue) for nuclei. Scale bar, 50 μm.

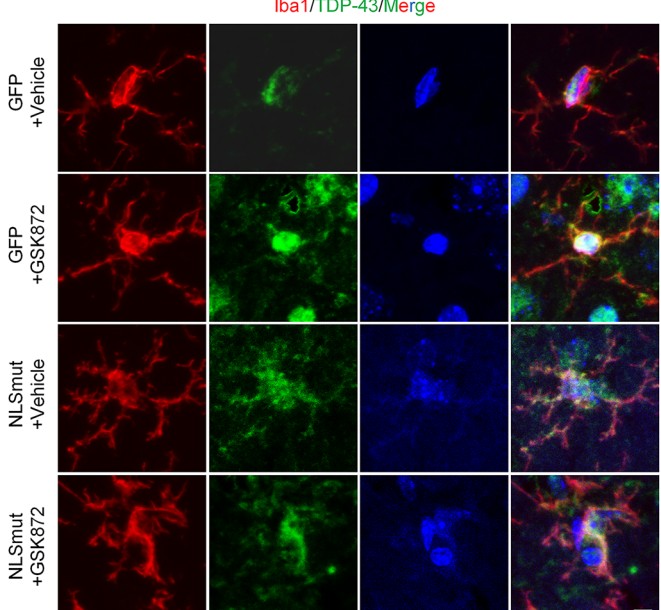

**Figure EV4.** (relative to Figs. 6 and 7) Immunofluorescence staining validated the overexpression of TDP-43 NLSmut in microglia in mice.

Immunofluorescence of TDP-43 (green) and Iba1 (red) in GFP control and TDP-43-NLSmut mice treated with or without GSK872. Hoechst (blue) for nuclei. Scale bar, 5 μm.

