## [Peer Review File · EMBO Molecular Medicine]

TDP-43 pathology triggers neuroinflammation and cognitive impairment by inducing microglial necroptosis

Shenrui Guo, Hongfu Jin, Hui Sun, Shuo Huang, Yuanyuan Chen, Yuge Chang, Yu Zhang, Lin Ding, Suyun Chen, Chenglai Fu, Yafu Yin, and Weiwei Cheng

Corresponding authors: Weiwei Cheng (chengweiwei8416@xinhumed.com.cn) , Chenglai Fu (fuchenglai@xinhumed.com.cn), Yafu Yin (yinyafu@shsmu.edu.cn)

Review Timeline:

Submission Date:	18th Oct 25
Editorial Decision:	11th Nov 25
Revision Received:	10th Jan 26
Editorial Decision:	28th Jan 26
Revision Received:	4th Feb 26
Accepted:	10th Feb 26

Editor: Zeljko Durdevic

Transaction Report:

11th Nov 2025

Dear Dr. Cheng,

Thank you for the submission of your manuscript to EMBO Molecular Medicine. We have now received feedback from the two reviewers who agreed to evaluate your manuscript. Both referees recognize interest of the study but also raise important and partially overlapping concerns that should be addressed in a major revision. If you would like to discuss further the points raised by the referees, I am available to do so via email or video. Let me know if you are interested in this option.

We would welcome the submission of a revised version within three months for further consideration. Please let us know if you require longer to complete the revision.

I look forward to receiving your revised manuscript.

Yours sincerely,

Zeljko Durdevic

Zeljko Durdevic
Senior Editor
EMBO Molecular Medicine

We require:

- 1) A .docx formatted version of the manuscript text (including legends for main figures, EV figures and tables). Please make sure that the changes are highlighted to be clearly visible.
- 2) Individual production quality figure files as .eps, .tif, .jpg (one file per figure). For guidance, download the 'Figure Guide PDF': (<https://www.embopress.org/page/journal/17574684/authorguide#figureformat>).
- 3) A .docx formatted letter INCLUDING the reviewers' reports and your detailed point-by-point responses to their comments. As part of the EMBO Press transparent editorial process, the point-by-point response is part of the Review Process File (RPF), which will be published alongside your paper.
- 4) A complete author checklist, which you can download from our author guidelines (<https://www.embopress.org/page/journal/17574684/authorguide#submissionofrevisions>). Please insert information in the checklist that is also reflected in the manuscript. The completed author checklist will also be part of the RPF.
- 5) Please note that all corresponding authors are required to supply an ORCID ID for their name upon submission of a revised manuscript.
- 6) It is mandatory to include a 'Data Availability' section after the Materials and Methods. Before submitting your revision, primary datasets produced in this study need to be deposited in an appropriate public database, and the accession numbers and

database listed under 'Data Availability'. Please remember to provide a reviewer password if the datasets are not yet public (see <https://www.embopress.org/page/journal/17574684/authorguide#dataavailability>).

12) Author contributions: You will be asked to provide CRediT (Contributor Role Taxonomy) terms in the submission system. These replace a narrative author contribution section in the manuscript.

13) A Conflict of Interest statement should be provided in the main text.

14) Every published paper now includes a 'Synopsis' to further enhance discoverability. Synopses are displayed on the journal webpage and are freely accessible to all readers. They include a short stand first (maximum of 300 characters, including space) as well as 2-5 one-sentences bullet points that summarizes the paper. Please write the bullet points to summarize the key NEW findings. They should be designed to be complementary to the abstract - i.e. not repeat the same text. We encourage inclusion of key acronyms and quantitative information (maximum of 30 words / bullet point). Please use the passive voice. Please attach these in a separate file or send them by email, we will incorporate them accordingly.

15) Include a Reagents and Tools Table as part of the Methods section, which can be downloaded from our author guidelines (<https://www.embopress.org/page/journal/17574684/authorguide#structuredmethods>)

***** Reviewer's comments *****

Referee #1 (Comments on Novelty/Model System for Author):

Microglial depletion of endogenous TDP-43 must be proven in NLSmut TDP-43 mice, which express mutated TDP-43 in neurons.

Referee #1 (Remarks for Author):

In their study entitled 'TDP-43 pathology triggers neuroinflammation and cognitive impairment by inducing microglial necroptosis', the authors used BV2 microglial cells stably expressing NLS-mutated TDP-43 resulting in its cytoplasmic accumulation. This induces necroptosis pathway activation which in turn leads to neuroinflammation, cell death and phagocytosis decrease. The authors found that induction of necroptosis relies on enhanced Ripk3 transcription due to the loss of nuclear TDP-43 associated repression.

Then, they used AAV injection with neuron-specific promoter to drive expression of NLSmut-TDP-43 in the mouse hippocampus and found that microglia exhibit increase expression of necroptotic markers. This was suppressed by treatment with the RIPK3 kinase inhibitor GSK872. Finally, they found that spatial memory was impaired in TDP-43 NLSmut mice and that GSK872 treatment mitigate this defect.

This study is interesting but lacks of western-blot quantification. In addition, some western blots are of poor quality, making them difficult to analyze. Also, the in vitro and in vivo experiments are not the same (microglial versus neuronal expression of NLSmut TDP-43), making difficult to draw strong conclusions. In particular, the nuclear depletion of endogenous microglial TDP-43 in NLSmutTDP43 mice was not shown. Below, you will find my concerns about this study.

Major remarks:

- Quantification analyses must be done for all western-blots experiments to reinforce author's conclusions.
- Fig.1A/Uncropped blots images of Fig.1A: the membrane used for TDP-43 labeling is not the same that the one used for LaminB1 and Tubulin staining. To be correctly analyzable, the loading control stainings must be done on the same membrane than the one used to detect the protein of interest. This remark is also true for the other blots shown in other figures.
- Why two bands are present in the tubulin condition (also the case in Uncropped blots images of Fig.1C but not in Uncropped blots images of Fig.3A or Supp 3.B) ?
- Fig.1B: Why is there a Myc staining in the GFP condition as only TDP-43 contains a myc tag ? Please add DAPI staining to visualize the cell nucleus and clearly show that endogenous nuclear TDP-43 levels are decreased. This must be quantified.
- The BV2-NLS mut cells showed cytoplasmic accumulation of TDP-43, which seems normal as TDP-43 is mutated in its nuclear localization signal. However, it does not seem to produce cytoplasmic aggregates. How the authors explain the nuclear depletion of endogenous TDP-43 without aggregate formation (generally, the endogenous protein is sequestered into cytoplasmic aggregates) ?
- Fig.1C: from this figure, it seems that TDP-43 expression levels are similar between GFP and NLSmut conditions. This is surprising as the NLSmut condition should over-express TDP-43 compared to GFP condition. Could the authors quantify the level of TDP-43NLSmut overexpression compared to control.
- Uncropped blots images of Fig.1C: why several membranes were used to show the truncated forms of TDP-43 as only one TDP-43 recognizing-antibody was used ? Please perform this western blot on the same membrane in order to estimate the quantity of each band relative to the others ?
- Please confirm the RIPK3 upregulation by performing immunostaining on BV2-NLSmut cells.
- From Fig.2F, it appears to me that p-MLKL is mostly localized in the nucleus of BV2-NLSmut cells but not at the membranes as authors said. Please clarify this.
- I do not understand why the RIPK3 profile is not the same between blots. Are the authors sure that the RIPK3 weak bands on Fig.3A correspond to RIPK3 and not the two strong bands below (that are absent in GFP condition)?
- Suppl Fig.6: Please do a high magnification to clearly show the nuclear depletion of TDP-43.
- In mice experiments, TDP-43NLSmut is targeted to neurons but not to microglia. As in BV2 cells, TDP-43NLSmut does not seem to aggregate in neurons. From the authors, RIPK3 upregulation in BV2-NLSmut cells is coming from the absence of nuclear TDP43, which release the transcriptional inhibition at RIPK3 gene locus. Consistently, it is expected that neuronal expression of TDP-43NLSmut must deplete endogenous TDP-43 from microglia. However, this was not shown. It could be possible that the effect of GSK872 was due to neuronal nuclear depletion of TDP-43 rather than microglial depletion.

- GSK872 inhibits the kinase activity of RIPK3. However, the authors indicate that GSK872 treatment of TDP-43NLSmut mice decrease RIPK3 expression. This is not consistent with their model that predict that nuclear depletion of NLSmutTDP-43 releases Ripk3 transcription inhibition, which should not be inhibited by GSK872.

Minor points:

- Page 1, Line 19: please add references showing that 'impaired phagocytic function can lead to the accumulation of neurotoxic aggregates and neuronal loss'.
- Fig 2C: k is missing in the name of Ripk3
- Page 16, lines 1 and 2: the 7 is missing in (Fig. E-F)
- The Fig. 7G is not cited into the text.
- Please discuss what could be the ligand(s) that activate necroptosis in BV2-NLSmut cells. Indeed, necroptosis relies on RIPK3 activation rather than overexpression.

Referee #2 (Comments on Novelty/Model System for Author):

The combination of the BV2 microglial cell line and the hippocampal injection mouse model is adequate for the initial testing of the proposed mechanism. The BV2 line allows for controlled mechanistic studies in vitro, while the mouse model demonstrates the relevance of the findings in a complex neural circuit. To further improve the model's relevance:

As the model involves viral overexpression of a mutant protein, it would be important to more thoroughly characterize the resulting pathology. This includes providing direct evidence (e.g., via fractionation assays or specific immunohistochemistry) for the presence of cytoplasmic TDP-43 in the mouse hippocampus, which is a cornerstone of the study's hypothesis.

Referee #2 (Remarks for Author):

Comments to the Author,

The study demonstrates that mislocalized cytoplasmic TDP-43 contributes to neuroinflammation and cognitive impairment by engaging RIPK3 in microglia. However, the following points require clarification and strengthening of the evidence: The Western blot results need quantitative analysis with statistical testing. Similarly, the number of cells in the immunofluorescence images (e.g., Fig. 6) should be quantified to demonstrate statistical differences. Following transfection with TDP-43 NLSmut, an increase in p-TDP-43 and TDP-43 CTF levels was observed in BV2 cells. Could the authors clarify whether these elevated species originate from the endogenous TDP-43 or the exogenous TDP-43 NLSmut? The mice model was established by hippocampal stereotactic injection of TDP-43 NLSmut virus to investigate RIPK3 function. Please explain the rationale for specifically targeting the hippocampus. Furthermore, a clearer characterization of this model is needed: what are its key pathological features, and is there direct evidence confirming the presence of cytoplasmic TDP-43? While the increased microglial number and morphological changes in the TDP-43 NLSmut mice model suggest neuroinflammation, this would be strengthened by measuring pro-inflammatory cytokines such as IL-1 β , IL-6, and TNF- α in the hippocampus.

We thank the editor for handling the review work of our manuscript entitled “TDP-43 pathology triggers neuroinflammation and cognitive impairment by inducing microglial necroptosis” (EMM-2025-22816) and thank the constructive comments raised by the reviewers. We have revised the manuscript accordingly and have also uploaded all the materials required by the journal. We believe the changes have significantly improved the manuscript.

Point by point response to the reviewers' comments

Reviewer's comments

Referee #1 (Comments on Novelty/Model System for Author):

Microglial depletion of endogenous TDP-43 must be proven in NLSmut TDP-43 mice, which express mutated TDP-43 in neurons.

Response: We thank the reviewer for the critical suggestion. In direct response, we performed the experiment of co-staining for Iba1 and TDP-43. The new data (Fig.EV4) clearly show nuclear decrease of endogenous TDP-43 in Iba1 positive microglial cells in TDP-43 NLSmut mice. This result has also been added in the Results section (Page 8, Line 6-7).

Referee #1 (Remarks for Author):

In their study untitled 'TDP-43 pathology triggers neuroinflammation and cognitive impairment by inducing microglial necroptosis', the authors used BV2 microglial cells stably expressing NLS-mutated TDP-43 resulting in its cytoplasmic accumulation. This induces necroptosis pathway activation which in turn leads to neuroinflammation, cell death and phagocytosis decrease. The authors found that induction of necroptosis relies on enhanced Ripk3 transcription due to the loss of nuclear TDP-43 associated repression. Then, they used AAV injection with neuron-specific promoter to drive expression of NLSmut-TDP-43 in the mouse hippocampus and found that microglia exhibit increase expression of necroptotic markers. This was suppressed by treatment with the RIPK3 kinase inhibitor GSK872. Finally, they found that spatial memory was impaired in TDP-43 NLSmut mice and that GSK872 treatment mitigate this defect.

This study is interesting but lacks of western-blot quantification. In addition, some western blots are of poor quality, making them difficult to analyze. Also, the in vitro and in vivo experiments are not the same (microglial versus neuronal expression of NLSmut TDP-43), making difficult to draw strong conclusions. In particular, the nuclear depletion of endogenous microglial TDP-43 in NLSmutTDP43 mice was not shown. Below, you will find my concerns about this study.

Major remarks:

Q1- Quantification analyses must be done for all western-blot experiments to reinforce author's conclusions.

Response: Following the reviewer's advice, we have now included quantitative analyses for all Western blots in figures. These results are displayed alongside the representative blots in the revised figures, consistently support and strengthen our original conclusions.

Q2- Fig.1A/Uncropped blots images of Fig.1A: the membrane used for TDP-43 labeling is not the same that the one used for LaminB1 and Tubulin staining. To be correctly analyzable, the loading control stainings must be done on the same membrane than the one used to detect the protein of interest. This remark is also true for the other blots shown in other figures.

Response: We sincerely appreciate the reviewer's thorough observations. We have now thoroughly revised the presentation of our uncropped Western blots with clearer annotations to explicitly show that the protein of interests and loading controls were detected on the same membrane. For the few instances where different membranes were originally used, we have added more repeats to meet the requirement. These new, rigorously controlled results have been used for quantification and displayed in the revised figures.

Q3- Why two bands are present in the tubulin condition (also the case in Uncropped blots images of Fig.1C but not in Uncropped blots images of Fig.3A or Supp 3.B)?

Response: We appreciate the reviewer for this observation. The lower molecular weight band is a residual signal from the prior incubation with the TDP-43 antibody. To avoid further confusion, we have now explicitly annotated the uncropped blots to distinguish the protein of interest signal from the internal control band.

Q4- Fig.1B: Why is there a Myc staining in the GFP condition as only TDP-43 contains a myc tag? Please add DAPI staining to visualize the cell nucleus and clearly show that endogenous nuclear TDP-43 levels are decreased. This must be quantified.

Response: Thank you for pointing out these points. We clarify that both the GFP and TDP-43 NLSmut constructs contain an N-terminal myc tag, which has been corrected in the manuscript. The nuclear decrease of TDP-43 was directly confirmed by quantitative WB analysis of subcellular fractions, presented in Fig.1A. The primary purpose of Fig.1B presentation is to show the succeed overexpression of myc-TDP-43 and the cytoplasmic localized myc-TDP-43. As suggested, we have added Hoechst staining in Fig.1B. To response to the reviewer, we also quantified the TDP-43 signal in the cytoplasm and nucleus and presented it in Fig.EV1.

Q5- The BV2-NLS mut cells showed cytoplasmic accumulation of TDP-43, which seems normal as TDP-43 is mutated in its nuclear localization signal. However, it does not seem to produce cytoplasmic aggregates. How the authors explain the nuclear depletion of

endogenous TDP-43 without aggregate formation (generally, the endogenous protein is sequestered into cytoplasmic aggregates) ?

Response: We thank the reviewer for this insightful comment. We agree the absence of cytoplasmic aggregates in our model. We interpret this phenotype as capturing an early, pre-aggregation phase of TDP-43 pathology. At this stage, the mis-localized TDP-43 is diffusely distributed in the cytoplasm but is already functionally disruptive, leading to a nuclear decrease in endogenous TDP-43 and transcriptional dysregulation. The lack of overt aggregates in our model is consistent with previous reports using similar TDP-43 NLSmut constructs (e.g., Winton MJ et al, 2008), supporting the biological relevance of this early pathogenic state.

Winton MJ, Igaz LM, Wong MM, Kwong LK, Trojanowski JQ, Lee VM. Disturbance of nuclear and cytoplasmic TAR DNA-binding protein (TDP-43) induces disease-like redistribution, sequestration, and aggregate formation. *J Biol Chem.* 2008 May 9;283(19):13302-9. doi: 10.1074/jbc.M800342200. Epub 2008 Feb 27.

Q6- Fig.1C: from this figure, it seems that TDP-43 expression levels are similar between GFP and NLSmut conditions. This is surprising as the NLSmut condition should over-express TDP-43 compared to GFP condition. Could the authors quantify the level of TDP-43NLSmut overexpression compared to control.

Response: We thank the reviewer for raising this important point. Quantitative analysis confirms that total TDP-43 level is not statistically significantly elevated in BV2-NLSmut cells compared to controls (Data was not shown in the manuscript). We interpret this result in light of the well-established autoregulatory property of TDP-43, which maintains its total cellular levels within a narrow homeostatic range, even under conditions of sustained overexpression.

Importantly, we can confirm the successful expression of the exogenous protein from two aspects:

1. Immunoblotting with an anti-myc antibody clearly detects the exogenous TDP-43 NLSmut protein. We have shown this result in the revised Fig.1C.
2. Despite the steady total level, we observe a significant and specific increase in pathological TDP-43 species, namely phosphorylated TDP-43 and C-terminal fragments, in BV2-NLSmut cells (Fig.1C).

Q7- Uncropped blots images of Fig.1C: why several membranes were used to show the truncated forms of TDP-43 as only one TDP-43 recognizing-antibody was used ? Please perform this western blot on the same membrane in order to estimate the quantity of each band relative to the others?

Response: We appreciate the reviewer's close attention to the details. To clarify, the multiple images shown originate from the same membrane. This antibody recognizes the C-terminal of TDP-43 and thus is able to detect the truncated proteins. We performed

multiple exposures to optimally visualize both the highly abundant full-length TDP-43 (~43 kDa) and the less abundant C-terminal fragments within a linear detection range. In addition, the membrane was physically trimmed to allow independent optimization for resolving the closely migrating bands at ~43 kDa and ~37 kDa. We have now provided comprehensively annotated uncropped images to eliminate any confusion and to confirm that all signals were derived from the same membrane.

Q8- Please confirm the RIPK3 upregulation by performing immunostaining on BV2-NLSmut cells.

Response: We have accordingly performed immunostaining on BV2-NLSmut cells. The results confirm RIPK3 upregulation, the according images are displayed in Fig.2E.

Q9- From Fig.2F, it appears to me that p-MLKL is mostly localized in the nucleus of BV2-NLSmut cells but not at the membranes as authors said. Please clarify this.

Response: We appreciate the reviewer's comment. We agree that majority of p-MLKL is not localized at the plasma membrane. Our key finding is the significant increase in p-MLKL at the plasma membrane in BV2-NLSmut cells compared to controls. To visualize this more clearly, we have now indicated the membrane-localized p-MLKL with arrows in the revised Fig.2G and Fig.3C.

While MLKL phosphorylation is widely described as a cytoplasmic event, the observed nuclear localization of p-MLKL in our model has been documented in other cellular contexts upon necroptotic activation (e.g., Zhang T et al, 2020.). The nuclear localization observed here may result from certain cellular contexts and reflect cell type-specific regulation.

Reference: Zhang T, Yin C, Boyd DF, Quarato G, Ingram JP, Shubina M, Ragan KB, Ishizuka T, Crawford JC, Tummers B, Rodriguez DA, Xue J, Peri S, Kaiser WJ, López CB, Xu Y, Upton JW, Thomas PG, Green DR, Balachandran S. Influenza Virus Z-RNAs Induce ZBP1-Mediated Necroptosis. *Cell*. 2020 Mar 19;180(6):1115-1129.e13.

Q10- I do not understand why the RIPK3 profile is not the same between blots. Are the authors sure that the RIPK3 weak bands on Fig.3A correspond to RIPK3 and not the two strong bands below (that are absent in GFP condition)?

Response: We thank the reviewer for this careful observation. The original blot for Fig.3A appears to have been an outlier. To ensure clarity and reproducibility, we have replaced that band with the other repeat in the revised Fig.3A, showing a clear and consistent RIPK3 band pattern that aligns with our other results (e.g. Fig.2D) and supports our conclusion.

Q11- Suppl Fig.6: Please do a high magnification to clearly show the nuclear depletion of TDP-43.

Response: We now have added the high-magnification images in Fig.EV3 to clearly

show the nuclear depletion of TDP-43.

Q12- In mice experiments, TDP-43NLSmut is targeted to neurons but not to microglia. As in BV2 cells, TDP-43NLSmut does not seem to aggregate in neurons. From the authors, RIPK3 upregulation in BV2-NLSmut cells is coming from the absence of nuclear TDP43, which releases the transcriptional inhibition at RIPK3 gene locus. Consistently, it is expected that neuronal expression of TDP-43NLSmut must deplete endogenous TDP-43 from microglia. However, this was not shown. It could be possible that the effect of GSK872 was due to neuronal nuclear depletion of TDP-43 rather than microglial depletion.

Response: We thank the reviewer for raising this critical point. In direct response, we performed the experiment of co-staining for Iba1 and TDP-43. The new data (Fig.EV4) clearly show nuclear decrease of endogenous TDP-43 in Iba1 positive microglial cells in TDP-43 NLSmut mice. This result has also been added in the Results section (Page 8, Line 6-7).

Although we cannot formally exclude a potential effect of GSK872 on neurons *in vivo*, our *in vitro* data provide direct and clear evidence that GSK872 rescues microglial functions by inhibiting necroptosis in microglia. The consistency between these cellular findings and the *in vivo* rescue effects on neuroinflammation, necroptosis markers expression in microglia cells and behavioral performance, strongly supports our primary conclusion that microglial necroptosis is a key driver of the observed pathology and a promising therapeutic target. Whether GSK872 may also exert direct effects on neurons in the intact brain remains an open question worthy of future study.

Q13- GSK872 inhibits the kinase activity of RIPK3. However, the authors indicate that GSK872 treatment of TDP-43NLSmut mice decreases RIPK3 expression. This is not consistent with their model that predicts that nuclear depletion of NLSmutTDP-43 releases Ripk3 transcription inhibition, which should not be inhibited by GSK872.

Response: We thank the reviewer for catching this error. Our data show that GSK872 inhibits the phosphorylation of RIPK3, but does not reduce its protein expression. We have corrected the manuscript to state clearly that GSK872 inhibits RIPK3 kinase activity (p-RIPK3) and downstream signaling, not its expression level (Page 8, Line 14-15).

Minor points:

- Page 1, Line 19: please add references showing that 'impaired phagocytic function can lead to the accumulation of neurotoxic aggregates and neuronal loss'.

Response: Thanks. The reference has been added (Page 1, Line 20-21).

- Fig 2C: k is missing in the name of Ripk3

Response: Thanks. It has been corrected.

- Page 16, lines 1 and 2: the 7 is missing in (Fig. E-F)

Response: Following the reorganization of figures, we have carefully checked and corrected all figure citations throughout the manuscript.

- The Fig.7G is not cited into the text.

Response: Thank you for pointing this out. Figure 7G is now presented as part of the Synopsis. It has been removed from the main figure set.

- Please discuss what could be the ligand(s) that activate necroptosis in BV2-NLSmut cells. Indeed, necroptosis relies on RIPK3 activation rather than overexpression.

Response: We appreciate the reviewer's question. The most probable ligand is autocrine TNF- α , which is significantly upregulated in BV2-NLSmut cells. This ligand can initiate the canonical necroptotic cascade via TNFR1-RIPK1. Additionally, the massive overexpression of RIPK3 likely sensitizes the pathway, allowing this autocrine signal to efficiently trigger necroptosis even at low thresholds. We have added this discussion to the manuscript (Page 10, Line 6-13).

Referee #2 (Comments on Novelty/Model System for Author):

The combination of the BV2 microglial cell line and the hippocampal injection mouse model is adequate for the initial testing of the proposed mechanism. The BV2 line allows for controlled mechanistic studies in vitro, while the mouse model demonstrates the relevance of the findings in a complex neural circuit. To further improve the model's relevance:

As the model involves viral overexpression of a mutant protein, it would be important to more thoroughly characterize the resulting pathology. This includes providing direct evidence (e.g., via fractionation assays or specific immunohistochemistry) for the presence of cytoplasmic TDP-43 in the mouse hippocampus, which is a cornerstone of the study's hypothesis.

Response: Thanks for nice words. We agree with the reviewer's comments. We have shown the presence of cytoplasmic TDP-43 in neurons in the mouse hippocampus. To clearly show this phenomenon, we presented with new images with higher resolution and higher magnification (Fig.EV3).

Referee #2 (Remarks for Author):

Comments to the Author,

The study demonstrates that mislocalized cytoplasmic TDP-43 contributes to neuroinflammation and cognitive impairment by engaging RIPK3 in microglia. However, the following points require clarification and strengthening of the evidence:

Q1. The Western blot results need quantitative analysis with statistical testing. Similarly, the number of cells in the immunofluorescence images (e.g., Fig.6) should be quantified to demonstrate statistical differences.

Response: Following the reviewer's advice, we have now included quantitative analyses for all Western blots in figures. These results are displayed alongside the representative blots in the revised figures, consistently support and strengthen our original conclusions.

Similarly, we have quantified the key immunofluorescence experiments. The quantitative results are now displayed alongside the representative images, and details including the number of cells and fields analyzed are provided in the figure legends.

Q2. Following transfection with TDP-43 NLSmut, an increase in p-TDP-43 and TDP-43 CTF levels was observed in BV2 cells. Could the authors clarify whether these elevated species originate from the endogenous TDP-43 or the exogenous TDP-43 NLSmut?

Response: We appreciate the reviewer's question regarding the origin of the pathological TDP-43 species. Due to the broad migration of CTF bands and the small size difference imparted by the Myc tag (~1 kDa), we cannot definitively assign whether CTFs derive from endogenous or exogenous TDP-43. The p-TDP-43 bands in BV2-NLSmut cells migrate at the same molecular weight as basal p-TDP-43 in controls but show markedly increased intensity, strongly suggesting they originate primarily from the endogenous TDP-43 pool.

Critically, the precise source of these species does not affect our conclusions, as the key finding is the accumulation of pathological TDP-43 forms and their functional consequences.

Q3. The mice model was established by hippocampal stereotactic injection of TDP-43 NLSmut virus to investigate RIPK3 function. Please explain the rationale for specifically targeting the hippocampus. Furthermore, a clearer characterization of this model is needed: what are its key pathological features, and is there direct evidence confirming the presence of cytoplasmic TDP-43?

Response: We thank the reviewer for these important questions regarding our mouse model. Our decision to target the hippocampus was based on well-established clinical evidence linking TDP-43 pathology in this region to cognitive impairment, particularly in conditions such as Limbic-predominant Age-related TDP-43 Encephalopathy (LATE). The hippocampus is an ideal region to investigate the cognitive deficits associated with TDP-43 pathology. This is directly supported by our observation of significant spatial memory impairments in the Morris Water Maze assay in TDP-43 NLSmut mice.

As presented in the manuscript and figures, our models have these key pathological features: microglial activation (significant increased Iba1 signal), cytoplasmic TDP-43 mis-localization in hippocampal neurons, upregulation of necroptosis markers and significant cognitive deficits. To response to the reviewer's request for clear characterization, we have now provided high quantity images with higher-resolution and higher-magnification. We have also added the rationale for specifically targeting the hippocampus in the revised manuscript in the result section (Page 7, Line 27; Page 8, Line 1-3).

Q4. While the increased microglial number and morphological changes in the TDP-43 NLSmut mice model suggest neuroinflammation, this would be strengthened by

measuring pro-inflammatory cytokines such as IL-1 β , IL-6, and TNF- α in the hippocampus.

Response: Thank you for the suggestion. We agree that our conclusions would be strengthened if the amount of IL-1 β , IL-6, and TNF- α is increased in TDP-43 NLSmut mice. Accordingly, we performed ELISA assay to detect the amount of IL-1 β , IL-6, and TNF- α in the hippocampal homogenates. However, the levels of the three pro-inflammatory cytokines were lower than detectable scope, we reasoned these cytokines are secreted by the microglia locally and function in a localized, paracrine manner, making their steady-state concentration in whole-tissue lysates undetectable by ELISA assays.

Critically, our study provides direct histological and functional evidence of neuroinflammation. We observed robust microgliosis with activated morphology in TDP-43 NLSmut mice. Treatment with the necroptosis inhibitor GSK872 significantly reduced microgliosis (the first row of low magnification images of Fig.6A), and this reduction correlated with the rescue of spatial memory deficits in the water maze test. These consistent cellular and behavioral findings strongly confirm the presence of functionally relevant neuroinflammation in our model.

28th Jan 2026

Dear Dr. Cheng,

Thank you for the submission of your revised manuscript to EMBO Molecular Medicine. I am pleased to inform you that we will be able to accept your manuscript pending the following final amendments:

- 1) Please address referee #1 remaining concern.
- 2) Source data: The uncropped blots document should be removed and included in the ZIPed source data files.
- 3) In the main manuscript file, please do the following:
 - Please address all comments suggested by our data editors listed below:
 - o Figure legends:
 1. Please note that the exact p values are not provided in the legends of figures 2F, 3E ,G; 4C, D; 5B, 7E, F; EV1 B, C; S1, S2.
 2. Please indicate the statistical test used for data analysis in the legend of figure 2A.
 3. Please note that information related to n is missing in the legends of figures 2A, 5A.
 4. Please note that the error bars are not defined in the legend of figure 5A.
 - Please add callouts Fig 7A, B.
 - In Methods, provide the antibody dilutions that were used for each antibody.
 - Remove Reagent Table from the manuscript text and upload it as a separate file.
 - Author contributions: Please remove it from the manuscript and specify author contributions in our submission system. CRediT has replaced the traditional author contributions section because it offers a systematic machine-readable author contributions format that allows for more effective research assessment. You are encouraged to use the free text boxes beneath each contributing author's name to add specific details on the author's contribution. More information is available in our guide to authors:
<https://www.embopress.org/page/journal/17574684/authorguide#authorshipguidelines>
- 4) Tables: Please remove Table EV1 and EV2 from the manuscript text and upload them with their legends as separate .xlsx files.
- 5) Appendix: Remove yellow highlights.
- 6) The Paper Explained: Rename "summary of the article for the non-informed reader" to The Paper Explained and add it to the main manuscript file.
- 7) Synopsis:
 - Synopsis image: Please resize the image to 550 px-wide x 300-600 pixels high.
 - Please check your synopsis text and image before submission with your revised manuscript. Please be aware that in the proof stage minor corrections only are allowed (e.g., typos).
- 8) As part of the EMBO Publications transparent editorial process (see our Editorial at <http://embomolmed.embopress.org/content/2/9/329>), EMBO Molecular Medicine will publish online a Review Process File (RPF) to accompany accepted manuscripts. This file will be published in conjunction with your paper and will include the anonymous referee reports, your point-by-point response and all pertinent correspondence relating to the manuscript. Let us know if you want to remove or not any figures from it prior to publication. Please note that the Authors checklist will be published at the end of the RPF.
- 9) Please provide a point-by-point letter INCLUDING my comments as well as the reviewer's reports and your detailed responses (as Word file).

I look forward to reading a new revised version of your manuscript as soon as possible.

Yours sincerely,

Zeljko Durdevic

Zeljko Durdevic
Senior Editor
EMBO Molecular Medicine

*** Instructions to submit your revised manuscript ***

*** PLEASE NOTE *** As part of the EMBO Publications transparent editorial process initiative (see our Editorial at <https://www.embopress.org/doi/pdf/10.1002/emmm.201000094>), EMBO Molecular Medicine will publish online a Review

Process File to accompany accepted manuscripts.

When preparing your revised manuscript, please refer to our guidelines: <https://link.springer.com/journal/44321/submission-guidelines#cms-Revised-submissions>. We perform an initial quality control of all revised manuscripts before re-review; failure to include requested items will delay the evaluation of your revision.

We require:

- 1) A .docx formatted version of the manuscript text (including legends for main figures, EV figures and tables). Please make sure that the changes are highlighted to be clearly visible.
- 2) Individual production quality figure files as .eps, .tif, .jpg (one file per figure). For guidance, download the 'Figure Guide PDF': <https://media.springernature.com/original/springer-cms/rest/v1/content/27825798/data/v1>.
- 3) A .docx formatted letter INCLUDING the reviewers' reports and your detailed point-by-point responses to their comments. As part of the EMBO Press transparent editorial process, the point-by-point response is part of the Review Process File (RPF), which will be published alongside your paper.
- 4) A complete author checklist, which you can download from our author guidelines. Please insert information in the checklist that is also reflected in the manuscript. The completed author checklist will also be part of the RPF.
- 5) Please note that all corresponding authors are required to supply an ORCID ID for their name upon submission of a revised manuscript.
- 6) It is mandatory to include a 'Data Availability' section after the Materials and Methods. Before submitting your revision, primary datasets produced in this study need to be deposited in an appropriate public database, and the accession numbers and database listed under 'Data Availability'. Please remember to provide a reviewer password if the datasets are not yet public.

- 7) For data quantification: please specify the name of the statistical test used to generate error bars and P values, the number (n) of independent experiments (specify technical or biological replicates) underlying each data point and the test used to calculate p-values in each figure legend. The figure legends should contain a basic description of n, P and the test applied. Graphs must include a description of the bars and the error bars (s.d., s.e.m.).
- 8) At EMBO Press we ask authors to provide source data for the main manuscript figures. You will receive a separate email with instructions for providing source data with your revised manuscript, including how to upload and organize the files.
- 9) Our journal encourages inclusion of *data citations in the reference list* to directly cite datasets that were re-used and obtained from public databases. Data citations in the article text are distinct from normal bibliographical citations and should directly link to the database records from which the data can be accessed. In the main text, data citations are formatted as follows: "Data ref: Smith et al, 2001" or "Data ref: NCBI Sequence Read Archive PRJNA342805, 2017". In the Reference list, data citations must be labeled with "[DATASET]". A data reference must provide the database name, accession number/identifiers and a resolvable link to the landing page from which the data can be accessed at the end of the reference.
- 10) We replaced Supplementary Information with Expanded View (EV) Figures and Tables that are collapsible/expandable online. A maximum of 5 EV Figures can be typeset. EV Figures should be cited as 'Figure EV1, Figure EV2' etc... in the text and their respective legends should be included in the main text after the legends of regular figures.

12) Author contributions: You will be asked to provide CRediT (Contributor Role Taxonomy) terms in the submission system. These replace a narrative author contribution section in the manuscript.

13) A Conflict of Interest statement should be provided in the main text.

14) Every published paper includes a 'Synopsis' to further enhance discoverability. Synopses are displayed on the journal webpage and are freely accessible to all readers. They include a short stand first (maximum of 300 characters, including space) as well as 2-5 one-sentences bullet points that summarizes the paper. Please write the bullet points to summarize the key NEW findings. They should be designed to be complementary to the abstract - i.e. not repeat the same text. We encourage inclusion of key acronyms and quantitative information (maximum of 30 words / bullet point). Please use the passive voice. Please attach these in a separate file or send them by email, we will incorporate them accordingly.

15) Include a Reagents and Tools Table as part of the Methods section, which can be downloaded from our author guidelines.

Graphs 800-1,200 DPI
Photos 400-800 DPI
Colour (only CMYK) 300-400 DPI"

*Additional important information regarding figures and illustrations can be found at
<https://media.springernature.com/original/springer-cms/rest/v1/content/27825798/data/v1>

***** Reviewer's comments *****

Referee #1 (Comments on Novelty/Model System for Author):

It is still unclear for me why the in vitro experiments rely on microglial expression of NLSmut TDP-43 while the in vivo experiments are based on neuronal expression of NLSmut TDP-43.

Referee #1 (Remarks for Author):

The authors responded appropriately to my comments.

Referee #2 (Remarks for Author):

Is suitable for publication.

Response to Referee#1

Referee #1 (Comments on Novelty/Model System for Author):

It is still unclear for me why the *in vitro* experiments rely on microglial expression of NLSmut TDP-43 while the *in vivo* experiments are based on neuronal expression of NLSmut TDP-43.

Response: We performed the *in vitro* experiments to study the mechanism of TDP-43 NLSmut-induced microglial dysfunction. In parallel, the *in vivo* model was designed to recapitulate the human disease scenario, in which TDP-43 pathology typically originates in neurons (Swanson MEV et al, 2025). Importantly, as shown in our new data (Fig EV4), neuronal TDP-43 pathology is sufficient to induce the key event—nuclear depletion of TDP-43 in microglia—which in turn activates the same necroptosis pathway identified *in vitro*. To ensure clarity for the reader, we have also emphasized this logical connection and the significance of our findings in the revised text (Page 7, Line 25-27; Page 8, Line 1-2).

Reference: Swanson MEV, Mrkela M, Turner C, Curtis MA, Faull RLM, Walker AK, Scotter EL. Neuronal TDP-43 aggregation drives changes in microglial morphology prior to immunophenotype in amyotrophic lateral sclerosis. *Acta Neuropathol Commun.* 2025 Feb 21;13(1):39.

10th Feb 2026

Dear Dr. Cheng,

We are pleased to inform you that your manuscript is accepted for publication and is now being sent to our publisher to be included in the next available issue of EMBO Molecular Medicine.

You may qualify for financial assistance for your publication charges - either via a Springer Nature fully open access agreement or an EMBO initiative. Check your eligibility: <https://link.springer.com/journal/44321/how-to-publish-with-us>

Zeljko Durdevic
Senior Editor
EMBO Molecular Medicine

>>> Please note that it is EMBO Molecular Medicine policy for the transcript of the editorial process (containing referee reports and your response letter) to be published as an online supplement to each paper. If you do NOT want this, you will need to inform the Editorial Office via email immediately. More information is available here: <https://link.springer.com/partners/embo-press/editorial-policies#Peer%20review>